# Symmetric Linear Bandits with Hidden Symmetry

**Nam Phuong Tran**
Department of Computer Science
University of Warwick
Coventry, United Kingdom
nam.p.tran@warwick.ac.uk

**The Anh Ta**
CSIRO's Data61
Marsfield, NSW, Australia
theanh.ta@csiro.au

**Debmalya Mandal**
Department of Computer Science
University of Warwick
Coventry, United Kingdom
debmalya.mandal@warwick.ac.uk

**Long Tran-Thanh**
Department of Computer Science
University of Warwick
Coventry, United Kingdom
long.tran-thanh@warwick.ac.uk

## Abstract

High-dimensional linear bandits with low-dimensional structure have received considerable attention in recent studies due to their practical significance. The most common structure in the literature is sparsity. However, it may not be available in practice. Symmetry, where the reward is invariant under certain groups of transformations on the set of arms, is another important inductive bias in the high-dimensional case that covers many standard structures, including sparsity. In this work, we study high-dimensional symmetric linear bandits where the symmetry is hidden from the learner, and the correct symmetry needs to be learned in an online setting. We examine the structure of a collection of hidden symmetry and provide a method based on model selection within the collection of low-dimensional subspaces. Our algorithm achieves a regret bound of $O(d_0^{2/3} T^{2/3} \log(d))$, where $d$ is the ambient dimension which is potentially very large, and $d_0$ is the dimension of the true low-dimensional subspace such that $d_0 \ll d$. With an extra assumption on well-separated models, we can further improve the regret to $O(d_0 \sqrt{T \log(d)})$.

## 1 Introduction

Stochastic bandit is a sequential decision-making problem in which a player, who aims to maximize her reward, selects an action at each step and receives a stochastic reward, drawn from an initially unknown distribution of the selected arm, in response. Linear stochastic bandit (LSB) [1] is an important variant in which the expected value of the reward is a linear function of the action. It is one of the most studied bandit variants and has many practical applications [27].

Actions in LSB are specified as feature vectors in $\mathbb{R}^d$ for very large feature dimension $d$, with performance i.e. the resulting regret scaling with $d$. Many works have addressed this curse of dimensionality by leveraging different low-dimensional structures as inductive biases for the learner. For example, sparsity, which assumes that the reward is a sparse linear function, has been used extensively in LSB to design bandit algorithms with better performance [2, 36, 23]. However, when the reward function lacks the structure for sparsity (which may occur in many real-world situations), a question arises: Are there different structures in the features of LSB that we can exploit to overcome the curse of dimensionality and design bandit algorithms with better performance?

In this paper, we study the inductive bias induced by symmetry structures in LSB, which is a more general model inductive bias than sparsity, and can facilitate efficient and effective learning [9]. Symmetry describes how, under certain transformations of the input of the problem, the outcome

38th Conference on Neural Information Processing Systems (NeurIPS 2024).

should either remain unchanged (*invariance*) or shift predictably (*equivariance*). In supervised learning, it has been empirically observed [17, 12] and theoretically proven [16, 6] that explicitly integrating symmetry into models leads to improved generalization error. However, in the literature on sequential decision-making, unlike sparsity, symmetry is rarely considered to date. This leads us to the following research question: Can one leverage *symmetry in sequential decision-making* tasks to enable effective exploration, and eventually break the curse of dimensionality?

In the machine learning literature, especially in supervised learning, most studies on symmetry assume prior knowledge of symmetry structures of the tasks under consideration [31, 16]. However, in numerous practical scenarios, the learner can only access to partial knowledge of the symmetry, necessitating the incorporation of symmetry learning mechanisms into the algorithms to achieve better performance. Examples of hidden symmetry can be found in multi-agent learning with cooperative behavior. As a motivating example, consider a company undertaking a large project that consists of several subtasks. The company must hire subcontractors with the goal of maximizing project quality while staying within budget constraints. Symmetry may arise in this situation when coalitions form among subcontractors, where members of a coalition work together to complete their allocated tasks using shared resources. In particular, the allocation of tasks within a coalition can be swapped without affecting overall team performance, inducing symmetry (i.e., performance remains invariant under permutation) in the task assignments. Coalitions among subcontractors often arise since sharing labor and resources reduces operational costs, making their work more efficient and cost-effective. However, these coalitions are typically *hidden* from the hiring company. One reason is that if the hiring company were aware of these collaborations, they could use this information to negotiate lower prices, knowing that the subcontractors are benefiting from shared resources. Another reason is that coalitions may raise concerns about collusion. In particular, in a competitive market, such as when subcontractors are hired through a bidding platform, coalition members can collaborate to manipulate the bidding process, which is considered unfair and could undermine the integrity of the bidding. For more practical examples of hidden symmetry in multi-agent reinforcement learning [34] and robotics [3], we refer the reader to Appendix D.2. Motivated by these examples, we believe that hidden symmetry is much more relevant in the context of sequential decision-making because the environment and its symmetry structure may not be readily available to the learner, as opposed to supervised learning and offline settings where data are provided during the training phase. As the learner has the power to freely collect data, it is expected that they will learn the hidden symmetry structures as they explore the environment.

Against this background, we ask the question of whether learner can leverage symmetry to enable effective exploration, and break the curse of dimensionality *without a prior knowledge* of the symmetry structure? Moreover, in the presence of symmetry, when can we design learning algorithms with optimal regret bounds? Towards answering this question, we investigate the setting of symmetric linear stochastic bandit in $\mathbb{R}^d$, where $d$ is potentially very large, and the expected reward function is invariant with respect to the actions of a *hidden* group $\mathcal{G}$ of coordinate permutations. Our contributions are summarised as follows:

1. We first give an impossibility result that *no algorithm can get any benefit by solely knowing that $\mathcal{G}$ is a subgroup of permutation matrices*. We achieve this by formally establishing a relation between the class of subgroup to the partition over the set $\{1, ..., d\}$. A direct implication of this impossibility result is that it is necessary to have further information about the structure of the hidden subgroup in order to achieve improved regret bounds.

2. Given this, we establish a cardinality condition on the class of symmetric linear bandits with hidden $\mathcal{G}$, in which the learner can learn the true symmetry structure and overcome the curse of dimensionality. Notably, this class includes a sparsity as a special case, and therefore inherits all the computational and statistical complexities of sparsity. Apart from sparsity, this class includes many other practically relevant classes, such as partitions that respect underlying hierarchies, non-crossing partitions, and non-nesting partitions (see Subsection 4.2.1 and Appendix D.2).

3. We cast our problem of learning with hidden subgroup $\mathcal{G}$ into model selection with collection of low-dimensional subspaces [25, 35]. To address the polynomial scaling of regret bounds with respect to the number of models and arms in previous works, we depart from model aggregation, which is typically used in LSB model selection, and introduce a new framework

inspired by Gaussian model selection [7] [1] and compressed sensing [8]. Based on this framework, we introduce a new algorithm, called EMC (for Explore Models then Commit). Under the assumption that the set of arm is exploratory, we prove that the regret bound of the EMC algorithm is $O(d_0^{2/3}T^{2/3}\log(d))$, and $O(d_0\sqrt{T}\log(d))$ with an additional assumption on well-separated partitions, where $d_0 \ll d$ is the dimension of the low-dimensional subspace associated with group $\mathcal{G}$.

To the best of our knowledge, our work is the first in the linear stochastic bandits literature that leverages symmetry in designing provably efficient algorithms. To save space, all proofs in this paper are deferred to the Appendix.

## 1.1 Related Work

We now briefly outline related work and compare them with our results. We refer the reader to Appendix F for a more in-depth literature review.

**Sparse linear bandits.** As we will explain in Section 4.2, sparsity is equivalent to a subset symmetry structures, and thus, can be seen as a special case of our setting. As such, we first review the literature of sparsity. Sparse linear bandits were first investigated in [2], where the authors achieve a regret of $\tilde{O}(\sqrt{dsT})$, with $\tilde{O}$ disregarding the logarithmic factor, and $s$ representing the sparsity level, and $T$ is the time horizon. This matches the regret lower bound for sparse bandits, which is $\Omega(\sqrt{dsT})$ [27]. More recently, the contextual version of linear bandits has gained popularity, where additional assumptions are made regarding the context distribution and set of arms [26, 36, 28, 11, 23] to avoid polynomial dependence on $d$. Notably, with the assumption on exploratory set of arms, [23] propose an Explore then Commit style strategy that achieves $\tilde{O}(s^{\frac{2}{3}}T^{\frac{2}{3}})$, nearly matching the regret lower bound $\Omega(s^{\frac{2}{3}}T^{\frac{2}{3}})$ [24] in the data-poor regime. As sparsity is equivalent to a subclass of hidden symmetry, all the lower bounds for sparse problems apply to our setting of learning with hidden symmetry.

**Model selection.** Our problem is also closely related to the problem of model selection in linear bandits, as the learner can collect potential candidates for the hidden symmetry model. Particularly, in model selection, there is a collection of $M$ features, and different linear bandits running with each of these features serve as base algorithms. By exploiting the fact that the data can be shared across all the base algorithms, the dependence of regret in terms of the number of features can be reduced to $\log(M)$. In particular, [25] propose a method that concatenates all $M$ features of dimension $d$ into one feature of dimension $Md$, and uses the Lasso estimation as a aggregation of models. Their algorithm achieves a regret bound of $O(T^{\frac{3}{4}}\sqrt{\log(M)})$ under the assumption that the Euclidean norm of the concatenated feature is bounded by a constant. However, in our case, the Euclidean norm of the concatenated feature vector can be as large as $\sqrt{M}$, which leads to a $\sqrt{M}$ multiplicative factor in the regret bound. Besides, [35] uses the online aggregation oracle approach, and is able to obtain regret of $O(\sqrt{KdT\log(M)})$, where $K$ is the number of arms. In contrast, *we use algorithmic mechanisms that are different from aggregation of models*. In particular, we explicitly exploit the structure of the model class as a collection of subspaces and invoke results from Gaussian model selection [21, 7] and dimension reduction on the union of subspaces [8]. With this technique, we are able to achieve $O(T^{\frac{2}{3}}\log(M))$, which is rate-optimal in the data-poor regime, has logarithmic dependence on $M$ without strong assumptions on the norm of concatenated features, and is independent of the number of arms $K$. We refer the reader to Section 4.2 for a more detailed explanation.

**Symmetry in online learning.** The notion of symmetry in Markov decision process dates back to works such as [22, 40]. Generally, the reward function and probability transition are preserved under an action of a group on the state-action space. Exploiting known symmetry has been shown to help achieve better performance empirically [43, 42] or tighter regret bounds theoretically [41]. However, all these works requires knowledge of symmetry group, while our setting consider hidden symmetry group which may be considerably harder. Hidden symmetry on the context or state space has been studied by few authors, with the term context-lumpable bandits [29], meaning that the set of contexts can be partitioned into classes of similar contexts. It is important to note that the symmetry group acts differently on the context space and the action space. As we shall explain in detail in Section 3, while one can achieve a reduction in terms of regret in the case of hidden symmetry acting on context

---

[1]We note that "Gaussian model selection" is a technique in statistics, similar to model aggregation (see [21]'s chapter 2 and 4), which should not be confused with "model selection" in the bandit literature.

spaces [29], this is not the case when the symmetry group acts on the action space. The work closest to ours is [39], where the authors consider the setting of a $K$-armed bandit, where the set of arms can be partitioned into groups with similar mean rewards, such that each group has at least $q > 2$ arms. With the constrained partition, the instance-dependent regret bounds are shown asymptotically to be of order $O\left(\frac{K}{q} \log T\right)$. Comparing to [39], we study the setting of stochastic linear bandits with similar arms, in which the (hidden) symmetry and linearity structure may intertwine, making the problem more sophisticated. We also impose different constraints on the way one partitions the set of arms, which is more natural in the setting of linear bandits with infinite arms.

## 2 Problem Setting

For any $k \in \mathbb{N}^+$, denote $[k] = \{1, \ldots, k\}$. For $\mathcal{X} \subseteq \mathbb{R}^d$, let $\Delta(\mathcal{X})$ denote the set of all probability measures supported on $\mathcal{X}$. Given a set $S \subset \mathbb{R}^k$, for some $k > 1$, denote $\Pi_S(x)$ as the Euclidean projection of $x \in \mathbb{R}^k$ on $S$, and $\mathrm{conv}(S)$ as the convex hull of $S$.

We denote by $T$ the number of rounds, which is assumed to be known in advance. Each round $t \in [T]$, the agent chooses an arm $x_t \in \mathcal{X} \subset \mathbb{R}^d$, and nature returns a stochastic reward $y_t = \langle x_t, \theta_\star \rangle + \eta_t$, where $\eta_t$ is an i.i.d. $\sigma$-Gaussian random variable. Now, denote $f(x_t) = \mathbb{E}[y_t \mid x_t]$. A bandit strategy is a decision rule for choosing an arm $x_t$ in round $t \in [T]$, given past observations up to round $t - 1$. Formally, a bandit strategy is a mapping $\mathcal{A} : (\mathcal{X} \times \mathbb{R})^T \to \Delta(\mathcal{X})$.

Let $x_\star = \arg\max_{x \in \mathcal{X}} f(x)$, and let $\mathbf{R}_T = \mathbb{E}\left[\sum_{t=1}^T \langle x_\star - x_t, \theta_\star \rangle\right]$ denote the expected cumulative regret. In this paper, we investigate the question whether one can get any reduction in term of regret, if the reward function is invariant under the action of a hidden group of transformations on the set of arms. We define the notion of group of symmetry as follows:

**Group and group action.** Given $d \in \mathbb{N}^+$, let $\mathcal{S}_d$ denote the symmetry group of $[d]$, that is, $\mathcal{S}_d := \{h : [d] \to [d] \mid h \text{ is bijective}\}$ the collection of all bijective mappings from $[d]$ to itself. We also define the group action $\hat{\phi}$ of $\mathcal{S}_d$ on the vector space $\mathbb{R}^d$ as

$$
\begin{aligned}
\phi : \mathcal{S}_d \times \mathbb{R}^d &\to \mathbb{R}^d \\
\left(g, (x_i)_{i \in [d]}\right) &\mapsto (x_{g(i)})_{i \in [d]}
\end{aligned}
\tag{1}
$$

In other words, a group element $g$ acts on an arm $x \in \mathbb{R}^d$ by permuting the coordinates of $x$. In the setting of linear bandit, the permutation group action also acts on the set of parameters via coordinate permutation. For brevity, we simply denote $g \cdot \theta$ and $g \cdot x$ as $\phi(g, \theta)$ and $\phi(g, x)$, respectively. Denote by $A_g$ the permutation matrix corresponding to $g$. We write $\mathcal{G} \leq \mathcal{S}_d$ to denote that $\mathcal{G}$ is a subgroup of $\mathcal{S}_d$. Given any point $\theta \in \mathbb{R}^d$, we write $\mathcal{G} \cdot \theta = \{g \cdot \theta \mid g \in \mathcal{G}\}$ to denote the orbit of $\theta$ under $\mathcal{G}$. It is well known that the orbit induced by the induced action of a subgroup $\mathcal{G} \leq \mathcal{S}_d$ corresponds to a set partition of $[d]$. We denote this partition as $\pi_\mathcal{G}$.

Let $\mathcal{G}$ be a subgroup of $\mathcal{S}_d$ that acts on $\mathbb{R}^d$ via the action $\phi$. In a symmetric linear bandit, the expected reward is invariant under the group action of $\mathcal{G}$ on $\mathcal{X}$, that is, $f(g \cdot x) = f(x)$. Due to the linear structure of $f$, this is equivalent to $g \cdot \theta_\star = \theta_\star$ for all $g \in \mathcal{G}$. We assume that, *while the group action $\phi$ is known to the learner, the specific subgroup $\mathcal{G}$ is hidden and must be learned in an online manner*.

## 3 Impossibility Result of Learning with General Hidden Subgroups

We now show how to frame the learning problem with hidden symmetry group as the problem of model selection. We further analyse the structure of the collection of models, and show that no algorithm can benefit by solely knowing that $\mathcal{G} \leq \mathcal{S}_d$, which implies that further assumptions are required to achieve significant improvement in term of regret.

### 3.1 Fixed Point Subspace and Partition

The analysis of learning with hidden subgroup requires a group-theoretic notion which is referred to as fixed-point subspaces [10]. As we shall explain promptly, there is a tight connection between the collection of fixed-point subspaces and set partitions.

**Fixed-point subspaces.** For a subset $\mathcal{X} \subseteq \mathbb{R}^d$, denote $\mathrm{Fix}_{\mathcal{G}}(\mathcal{X}) := \{x \in \mathcal{X} \mid g \cdot x = x, \ \forall g \in \mathcal{G}\}$ as the fixed-point subspace of $\mathcal{G}$; and $\mathcal{F}_{\mathcal{S}_d}(\mathcal{X}) := \{\mathrm{Fix}_{\mathcal{G}}(\mathcal{X}) \mid \mathcal{G} \leq \mathcal{S}_d\}$ as the collection of all fixed-point subspaces of all subgroups of $\mathcal{S}_d$. We simply write $\mathcal{F}_{\mathcal{S}_d} = \mathcal{F}_{\mathcal{S}_d}(\mathbb{R}^d)$ and $\mathrm{Fix}_{\mathcal{G}} = \mathrm{Fix}_{\mathcal{G}}(\mathbb{R}^d)$ for brevity.

**Set partition.** Given $d \in \mathbb{N}^+$, we denote $\mathcal{P}_d$ as the set of all partitions of $[d]$. Let $\mathcal{P}_{d,k}$ as the set of all partitions of $[d]$ with exactly $k$ classes, and $\mathcal{P}_{d,\leq k}$ be the set of all partitions of $[d]$ with at most $k$ classes. The number of set partitions with $k$ classes $|\mathcal{P}_{d,k}|$ is known as the Stirling number of the second kind, and $|\mathcal{P}_d|$ is known as Bell number.

### 3.2 Impossibility Result

**Problem with known symmetry.** Before discussing the problem of hidden symmetry, let us explain why the learner with an exact knowledge of $\mathcal{G}$ can trivially achieve smaller regret. The reason is that $\theta_\star \in \mathrm{Fix}_{\mathcal{G}}$ by the assumption that $\theta_\star$ is invariant w.r.t the action of group $\mathcal{G}$. If $\mathcal{G}$ is known in advance, the learner can restrict the support of $\theta_\star$ in $\mathrm{Fix}_{\mathcal{G}}$, and immediately obtains that the regret scales with $\dim(\mathrm{Fix}_{\mathcal{G}})$ instead of $d$, which can be significantly smaller (e.g., if $\mathcal{G} = \mathcal{S}_d$, then $\dim(\mathrm{Fix}_{\mathcal{G}}) = 1$).

For any subgroup, there exists a fixed point subspace, and some subgroups may share the same fixed point subspace. Therefore, instead of constructing a collection of subgroups, one can create a smaller collection of models using the collection of fixed point subspaces. As $\mathcal{G}$ is hidden, one must learn $\mathrm{Fix}_{\mathcal{G}}$ within the set of candidates $\mathcal{F}_{\mathcal{S}_d}$, leading to the formulation of the model selection.

**From the setting with hidden subgroup to the setting with hidden set partition.** Now, we discuss the structure of the collection of models $\mathcal{F}_{\mathcal{S}_d}$. First, we show the equivalent structure between the collection of fixed point subspaces and the set partitions as follows.

**Proposition 1.** *There is a bijection* $\mathbf{H}$ *between* $\mathcal{P}_d$ *and* $\mathcal{F}_{\mathcal{S}_d}$.

As there is a bijection between $\mathcal{P}_d$ and $\mathcal{F}_{\mathcal{S}_d}$, we can count the number of subspaces of each dimension $k$ explicitly using the following.

**Proposition 2** ([10]'s Theorem 14). *Given a subgroup* $\Gamma \leq \mathcal{S}_d$ *and its fixed-point subspace* $\mathrm{Fix}_\Gamma$, *suppose that* $\pi_\Gamma$ *partitions* $[d]$ *into* $k$ *classes, then* $\dim(\mathrm{Fix}_\Gamma) = k$.

By Proposition 2, we have that the number of subspaces of dimension $k$ in $\mathcal{F}_{\mathcal{S}_d}$ is exactly the number of set partitions with $k$-classes. Suppose that the learner knows that the orbit under action of $\mathcal{G}$ partitions the index of $\theta_\star$ into 2 equivalent classes that is, $\dim(\mathrm{Fix}_{\mathcal{G}}) = 2$. The learner cannot get any reduction in terms of regret.

**Proposition 3.** *Assume that the action set is the unit cube* $\mathcal{X} = \{x \in \mathbb{R}^d \mid \|x\|_\infty \leq 1\}$, *and* $f$ *is invariant w.r.t. action of subgroup* $\mathcal{G} \leq \mathcal{S}_d$, *such that* $\dim(\mathrm{Fix}_{\mathcal{G}}) = 2$. *Then, the regret of any bandit algorithm is lower bounded by* $\mathbf{R}_T = \Omega(d\sqrt{T})$.

The implication of Proposition 3 is that even if the learner knows $\theta_\star$ lies in an extremely low-dimensional subspace within the finite pools of candidates, they *still suffer a regret that scales linearly with the ambient dimension* $d$. This suggests that *further information about the group* $\mathcal{G}$ *must assumed to be known* in order to break this polynomial dependence on $d$ in the regret bound.

## 4 The Case of Hidden Subgroups with Subexponential Size

As indicated by Proposition 3, there is no improvement in terms of regret, despite the learner having access to a collection of extremely low-dimensional fixed point subspaces. Therefore, we assume that the learner can access only a reasonably small subset of the collection of low-dimensional fixed point subspaces. Let $d_0$ be the upper bound for the dimension of fixed point subspaces; that is, we know that the orbit of $\mathcal{G}$ partitions $[d]$ into at most $d_0$ classes. Now, let us assume that the learner knows that $\mathcal{G}$ does not partition $[d]$ freely, but must satisfy certain constraints, that is, $\pi_{\mathcal{G}} \in \mathcal{Q}_{d,\leq d_0} \subset \mathcal{P}_{d,\leq d_0}$. Here, $\mathcal{Q}_{d,\leq d_0}$ is a small collection of partitions with at most $d_0$ classes, which encodes the constraints on the way $\mathcal{G}$ partitions $[d]$. We introduce an assumption regarding the cardinality of $\mathcal{Q}_{d,\leq d_0}$, which is formally stated in Section 4.2. Using the Proposition 1, we can define the collection of fixed point subspaces associated with the collection of partition $\mathcal{Q}_{d,\leq d_0}$ via the bijection $\mathbf{H}$ as

$$\mathcal{M} := \mathbf{H}\left(\mathcal{Q}_{d,\leq d_0}\right) \quad \text{and} \quad M := |\mathcal{M}|.$$

In addition, let us define the extension of the collection $\mathcal{M}$ as $\overline{\mathcal{M}} := \{\mathrm{conv}\,(m \cup m') \mid m, \ m' \in \mathcal{M}\}$, where $\mathrm{conv}(S)$ is the convex hull of the set $S \subset \mathbb{R}^n$.

We have that $\overline{\mathcal{M}}$ is a collection of subspaces, that is, $\mathrm{conv}\,(m \cup m')$ is indeed a subspace [8]. Denote $\overline{M} := |\overline{\mathcal{M}}|$, then we have $\overline{M} = (M^2 - M)/2$. Moreover, if dimension of subspace in $\mathcal{M}$ is at most $d_0$, then the dimension of subspace in $\overline{\mathcal{M}}$ is at most $2d_0$.

## 4.1 The `Explore-Models-then-Commit` Algorithm

Given some $n \in [T]$, we define

$$Y = X\theta_\star + \boldsymbol{\eta}, \tag{2}$$

where $Y \in \mathbb{R}^n$, $X = [x_1, \ldots, x_n]^\top \in \mathbb{R}^{n \times d}$ is the design matrix, $\theta_\star \in \mathbb{R}^d$ is the true model; $\boldsymbol{\eta} = [\eta_1, ..., \eta_n]$. We have the information that $\theta_\star$ must be contained in some (not necessarily unique) subspace $m \in \mathcal{M}$. Denote by $d_m$ the dimension of $m$, we have $d_m \leq d_0$ for any $m \in \mathcal{M}$. Let $X_m = [\Pi_m(x_t)]_{t \in [n]}^\top$, and $S_m$ be the column space of $X_m$, one has $\dim(S_m) \leq d_m$. For any $m \in \mathcal{M}$, and given $Y$, let $\Pi_{S_m}(\cdot)$ be the projection onto $S_m$. Define

$$\widehat{\boldsymbol{f}}_m := \Pi_{S_m}(Y); \quad \widehat{\theta}_m := \arg\min_{\theta \in m} \|Y - X\theta\|_2^2. \tag{3}$$

Now, given $n$ data points, we can choose the model $\widehat{m} \in \mathcal{M}$ that minimises the least square error

$$\widehat{m} \in \arg\min_{m \in \mathcal{M}} \|Y - \widehat{\boldsymbol{f}}_m\|_2^2. \tag{4}$$

Based on the framework of model selection, we now introduce our Algorithm 1, `Explore-Models-then-Commit` (EMC). Our algorithm falls into the class explore-then-commit bandit algorithms. The exploration phase consists of $t_1$ rounds. During this phase, one samples data independently and identically distributed (i.i.d.) from an exploratory distribution $\nu$. After the exploration phase, one computes the solution to the model selection problem and then commits to the best arm corresponding to the chosen model.

**Remark 4.** The key step of Algorithm 1 that may incur significant costs is solving equation (4) (line 6). Without additional information about $\mathcal{M}$, one might need to enumerate all models in $\mathcal{M}$ and optimize among them, which would induce a time complexity of $O(nd^{cd_0})$. However, if we have more information about the partitions, e.g., if they are non-crossing or non-nesting partitions, their lattice structures can be exploited to speed up the optimization process of solving equation (4). Due to space limitations, we refer readers to Appendix D.3 for a detailed explanation of a subroutine that leverages these lattice structures for more efficient computation. Additionally, Section 6 demonstrates that our Algorithm 1, when using the lattice search algorithm for non-crossing partitions and non-nesting partitions as a subroutine, achieves polynomial computational complexity of $O(nd^5)$ and guarantees low regret.

## 4.2 Regret Analysis

The regret analysis of Algorithm 1 uses results from the Gaussian model selection literature [7, 21] as a basis. As such, we first state the assumptions that are common in the Gaussian model selection literature on the collection of models $\mathcal{M}$ and the set of arms $\mathcal{X}$ (Section 4.2.1). We then provide our main analysis in Section 4.2.2, highlighting the key technical novelties of our approach.

### 4.2.1 Assumptions

Recall that due to our lower bound in Proposition 3, further assumptions are required on the collection of fixed-point subspaces to achieve a reduction in terms of regret. As suggested by the model selection literature [25, 35], one can achieve regret in terms of $\log(M)$ for a collection of $M$ models. Adopting this idea, we make the following assumption regarding the number of potential fixed-point subspaces and the set of arms.

---

**Algorithm 1** Explore Models then Commit

1: Input: $T$, $\nu$, $t_1$
2: **for** $t = 1, \ldots, t_1$ **do**
3:     Independently pull arm $x_t$ according to $\nu$ and receive a reward $y_t$.
4: **end for**
5: $X \leftarrow [x_1, ..., x_{t_1}]^\top$, $Y \leftarrow [y_t]_{t \in [t_1]}$.
6: Compute $\widehat{m}$ as (4).
7: Compute $\widehat{\theta}_{t_1}$ as (3) corresponding to $\widehat{m}$.
8: **for** $t = t_1 + 1$ to $T$ **do**
9:     Take greedy actions:

$$x_t = \arg\min_{x \in \mathcal{X}} \left\langle \widehat{\theta}_{t_1}, x \right\rangle.$$

10: **end for**

---

**Assumption 5** (**Sub-exponential number of partitions**). *The partition corresponding to $\mathcal{G}$ belongs to a small subclass of partitions $\mathcal{Q}_{d,\leq d_0} \subset \mathcal{P}_{d,\leq d_0}$. In particular, $\pi_{\mathcal{G}} \in \mathcal{Q}_{d,d_\star}$, for some $d_\star \leq d_0$, and for each $k \in [d_0]$, there exists a constant $c > 0$, such that $|\mathcal{Q}_{d,k}| \leq O(d^{ck})$.*

**Assumption 6** (**Bounded set of arms**). *There are positive numbers $K_x, R_{\max}$, such that, for all $x \in \mathcal{X}$ and $m \in \overline{\mathcal{M}}$, $\|\Pi_m(x)\|_2^2 \leq K_x$, and $|\langle x, \theta_\star \rangle| \leq R_{\max}$.*

As a consequence of Assumption 5, the cardinality of the collection of fixed point subspaces is not too large, particularly, $M = O(d^{cd_0})$. First, we note that this class includes interval partitions, a structure equivalent to sparsity as a *strict* subset, as explained below.

**Remark 7** (**Equivalence between sparsity and interval partition**). A set partition of $[d]$ is an interval partition or partition of interval if its parts are interval. We denote $\mathcal{I}_d$ as the collection of all interval partition of $d$. $\mathcal{I}_d$ admits a Boolean lattice of order $2^{d-1}$, making it equivalent to the sparsity structure in $d - 1$ dimensions. Specifically, consider the set of entries of parameters $\varphi \in \mathbb{R}^d$ with a linear order, that is, $\varphi_1 \geq \varphi_2 \geq \cdots \geq \varphi_d$. Then define the variable $\theta \in \mathbb{R}^{d-1}$ such that $\theta_i = (\varphi_i - \varphi_{i+1})$. Each interval partition on the entries of $\varphi$ will determine a unique sparse pattern of $\theta$. Therefore, it is clear that the cardinality of the set of interval partition with $d_0$ classes is bounded as $|\mathcal{I}_{d,\leq d_0}| = O(d^{d_0})$. Moreover, as a result, symmetric linear bandit is strictly harder than sparse bandit and inherits all the computational complexity challenges of sparse linear bandit, including the *NP-hardness* of computational complexity.

Apart from sparsity, class of partitions with sub-exponential size also naturally appears when there is a hierarchical structure on the set $[d]$, and the partitioning needs to respect this hierarchical structure. A partition that respects an ordered tree groups the children of the same node into a single equivalence class, for example, see Figure 1. It is shown in [15] and the cardinality of the set of partitions that respect ordered trees is sub-exponential. Furthermore, as shown in [15] there is a bijection between partitions that respect ordered trees and the set of non-crossing partitions.

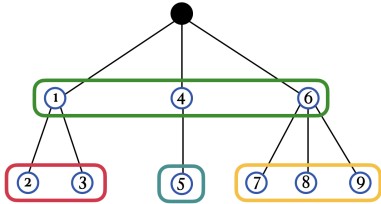

**Figure 1:** Partition that respects the underlying ordered tree.

A real-life example that meets these assumptions is the subcontractor example in the introduction: A hierarchical structure may exist, where a hired subcontractor can further subcontract parts of the work to others. A tree represents the hierarchical order among subcontractors, where subcontractors hired by another contractor can be grouped into one class. Further real-life examples of non-crossing partitions and other structured partitions that satisfy sub-exponential cardinality, such as non-nesting partitions and pattern-avoidance partitions [33], can be found in Appendix D.2.

Next, similar as [23], we define the exploratory distribution as follows.

**Definition 8** (**Exploratory distribution**). The exploratory distribution $\nu \in \Delta(\mathcal{X})$ is solution of the following optimisation problem

$$\nu = \underset{\omega \in \Delta(\mathcal{X})}{\arg\max} \lambda_{\min}\left(\mathbb{E}_{x \sim \omega}[xx^\top]\right), \quad V := \mathbb{E}_{x \sim \nu}[xx^\top], \quad C_{\min}(\mathcal{X}) := \lambda_{\min}(V). \quad (5)$$

Since our setting includes sparsity as a special case, the regret lower bound in [24] applies to our setting as well. In particular, we have:

**Proposition 9** (**Regret lower bound**). *There exist symmetric linear bandit instances in which Assumption 5, 6 hold with $K_x = 8d_0$, such that, any bandit algorithm must suffer regret $\mathbf{R}_T = \Omega\left(\min\left(C_{\min}(\mathcal{X})^{-\frac{1}{3}} d_0^{\frac{2}{3}} T^{\frac{2}{3}}, \sqrt{dT}\right)\right)$.*

We note that the lower bound can be relaxed if we have a stronger assumption on the group $\mathcal{G}$, which allows algorithm to go beyond the lower bound in Proposition 9 of the sparsity class. For example, if we consider a collection of fixed-point subspaces with a nested structure, similar to those discussed in [20], the algorithm may achieve a $O(d_0\sqrt{T})$ rate. The key takeaway is that symmetry exhibits significant flexibility in structure. Depending on the specific class of symmetry, one may achieve either no reduction at all or significant reduction in terms of regret bound.

### 4.2.2 Main Regret Upper Bound Result

We now state the main result of the regret upper bound for Algorithm 1.

**Theorem 10** (**Regret upper bound**). *Suppose the Assumptions 5, 6 hold. With the choice of* $t_1 = R_{\max}^{-\frac{2}{3}} \sigma^{\frac{2}{3}} C_{\min}^{-\frac{1}{3}}(\mathcal{X}) K_x^{\frac{1}{3}} d_0^{\frac{1}{3}} T^{\frac{2}{3}} (\log(dT))^{\frac{1}{3}}$, *then the regret of Algorithm 1 is upper bounded as*

$$\mathbf{R}_T = O\left(R_{\max}^{\frac{1}{3}} \sigma^{\frac{2}{3}} C_{\min}^{-\frac{1}{3}}(\mathcal{X}) K_x^{\frac{1}{3}} d_0^{\frac{1}{3}} T^{\frac{2}{3}} (\log(dT))^{\frac{1}{3}}\right) \tag{6}$$

**Remark 11.** We note that when $K_x = O(d_0)$, as in the lower bound instances, our upper bound is $\tilde{O}\left(C_{\min}^{-\frac{1}{3}}(\mathcal{X}) d_0^{\frac{2}{3}} T^{\frac{2}{3}}\right)$, which matches the lower bound in Proposition 9.

The main idea is to bound the risk error after exploration rounds, as stated in the following lemma which implies the regret bound after standard manipulations.

**Lemma 12.** *Suppose the Assumptions 5, 6 hold. For* $t_1 = \Omega(K_x^2 d_0 C_{\min}^{-2}(\mathcal{X}) \log(d/\delta))$, *with probability at least* $1 - \delta$, *one has the estimate*

$$\left\|\theta_\star - \widehat{\theta}_{t_1}\right\|_2 = O\left(\sqrt{\frac{\sigma^2 d_0 \log(d/\delta)}{C_{\min}(\mathcal{X}) t_1}}\right). \tag{7}$$

**Remark 13** (**Non-triviality of Lemma 12**). At the first glance, it seems that we can cast the problem of learning with a collection of $M$ subspaces into a model selection problem in linear bandit with $M$ features. This leads to a question: *Can we apply the model selection framework based on model aggregation in [35, 25] to our case?*

First, let us explain how to cast our problem into a model selection problem in linear bandit. For each subspace $m$, let $\Phi_m : \mathbb{R}^d \to \mathbb{R}^{d_0}$ be the feature map that computes the image of the projection $\Pi_m$ with respect to the orthogonal basis of subspace $m$. Thus, we then have a collection of $M$ features $\{\Phi_m\}_{m \in \mathcal{M}}$. Consider the algorithm introduced in [25], which concatenates the feature maps into $\Phi(x) = [\Phi_1(x), \ldots, \Phi_M(x)] \in \mathbb{R}^{Md_0}$, and the regret bound depends on $\|\Phi(x)\|_2$.

However, in our case where $\|\Phi_m(x)\|_2 < 1$, we can only bound $\|\Phi(x)\|_2 \leq \sqrt{M}$, which leads to a $\sqrt{M}$ dependence on regret, if we use their algorithm. Regarding [35], their algorithm aggregates the predictions among models for each arm, and based on that, they compute the distribution for choosing each arm. This leads to the regret scaling with the number of arms $K$, which is not feasible in our case when $K = \infty$.

We note that the similarity with the model selection technique in [25, 35] is that they use model aggregation among $\mathcal{M}$ to bound the prediction error $\sum_{t=1}^T \left\langle x_t, \widehat{\theta} - \theta_\star \right\rangle^2$, but this does not necessarily guarantee the risk error $\|\widehat{\theta} - \theta_\star\|_2$. The reason is that, although model aggregation can guarantee a small prediction error, it imposes no restriction on the estimator $\widehat{\theta}$, which limits its ability to leverage the further benign property of designed matrix $X$. Instead of model aggregation, our algorithm explicitly picks the best model from the pool $\mathcal{M}$, ensuring the following two properties: (1) The prediction error is small, similar to model aggregation; and (2) We can guarantee that $\widehat{\theta}$ lies in one of the subspaces of $\mathcal{M}$. The second property gives us control over $(\widehat{\theta} - \theta_\star)$ by ensuring it lies in at most $M^2$ subspaces. Then, exploiting the restricted isometry property (see Definition 15) of designed matrix $X$, we can guarantee that with $O(\log(M))$ exploratory samples, we can bound the risk error $\|\widehat{\theta} - \theta_\star\|_2$. This is crucial for eliminating polynomial dependence on $M$ and the number of arms $K$.

**Proof sketch of Lemma 12.** We provide a proof sketch here and defer their full proof to Appendix B.1. Our proof borrows techniques from Gaussian model selection [21] and the compressed sensing literature [8]. There are two steps to bound the risk error as in Lemma 12:

**Step 1 - Bounding the prediction error.** We can bound the prediction error $\left\|X\theta_\star - X\widehat{\theta}_{t_1}\right\|_2$ using the Gaussian model selection technique [21] as follows.

**Proposition 14.** *Let* $\boldsymbol{f}_\star = X\theta_\star$. *For the choice of* $\widehat{\boldsymbol{f}}_{\widehat{m}}$ *as in Eqn.* (3) *& Eqn.* (4), *with probability at least* $1 - \delta$, *there exists a constant* $C > 1$ *such that*

$$\left\|\widehat{\boldsymbol{f}}_{\widehat{m}} - \boldsymbol{f}_\star\right\|_2^2 \leq C\sigma^2 \log\left(M\delta^{-1}\right). \tag{8}$$

**Step 2 - Bounding the risk error from prediction error.** To bound the risk error from the prediction error, we invoke the restricted isometry property on the union of subspaces of a sub-Gaussian random matrix as in [8]. Note that $\widehat{\theta}$ and $\theta_\star$ can belong to two different subspaces of $\mathcal{M}$, and $\widehat{\theta} - \theta_\star$ may not lie in any subspace of $\mathcal{M}$, but in $\overline{\mathcal{M}}$. An important property of the design matrix $X$, which allows one to recover $\theta_\star$ with the knowledge that $\theta_\star$ is in a subspace $m \in \mathcal{M}$, can be captured by the following notion of restricted isometry property (RIP):

**Definition 15** (**Restricted isometry property**). For any matrix $X$, any collection of subspaces $\overline{\mathcal{M}}$ and any $\theta \in m \in \overline{\mathcal{M}}$, we define $\overline{\mathcal{M}}$-restricted isometry constant $\delta_{\overline{\mathcal{M}}}(X)$ to be the smallest quantity such that

$$(1 - \delta_{\overline{\mathcal{M}}}(X))\|\theta\|_2^2 \leq \|X\theta\|_2^2 \leq (1 + \delta_{\overline{\mathcal{M}}}(X))\|\theta\|_2^2. \tag{9}$$

We have the minimum number of samples required so that a random matrix $X$ satisfies RIP for a given constant with high probability as Proposition 16 below. Then, Lemma 12 is followed by combining Proposition 16 and Proposition 14.

**Proposition 16.** *Let* $X = [x_t]_{t \in [n]}$, *where* $x_t$*'s are is i.i.d. drawn from* $\nu$, *and let* $n = \Omega\left(C_{\min}^{-2}(\mathcal{X})K_x^2\left(\log(2\overline{M}d_0\delta^{-1})\right)\right)$. *Then, with probability at least* $1 - \delta$, *and for any* $\theta_1, \theta_2$ *in subspaces of* $\mathcal{M}$, *one has that*

$$\|\theta_1 - \theta_2\|_2^2 \leq 2C_{\min}^{-1}(\mathcal{X})n^{-1}\|X(\theta_1 - \theta_2)\|_2^2. \tag{10}$$

# 5 Improved Regret Upper Bound with Well-Separated Partitions

We now show that by adding more structure (i.e., well-separatedness) to the setting, we can further improve the regret upper bound to $O(\sqrt{T})$. In particular, we will introduce the notion of well-separatedness in this section, and show that this notion can lead to improved (i.e., $O(\sqrt{T})$) regret bounds.

For each partition $p \in \mathcal{P}_d$, there is a unique equivalence relation on $[d]$ corresponding to $p$. Denote by $\overset{p}{\sim}$ the equivalence relation corresponding to $p$. Next, we define well-separatedness.

**Assumption 17** (**Well-separated partitioning**). *Given the true subgroup* $\mathcal{G}$, *and the corresponding partition* $\pi_{\mathcal{G}}$. *For all* $(i, j)$ *such that* $i \overset{\pi_{\mathcal{G}}}{\not\sim} j$, *it holds that* $|\theta_{\star,i} - \theta_{\star,j}| \geq \varepsilon_0$, *for some* $\varepsilon_0 > 0$.

The implication of Assumption 17 is that the projection of $\theta_\star$ to any subspace $m \in \mathcal{M}$ not containing $\theta_\star$ will cause some bias in the estimation error. In particular, one can show that for any $m \in \mathcal{M}$ such that $\theta_\star \notin m$, it holds that

---

**Algorithm 2** Exploring Model then Commit with well-separated partition

---

1: Input: $T$, $\nu$, $t_2$
2: **for** $t = 1, \cdots, t_2$ **do**
3:     Independently pull arm $x_t$ according to $\nu$ and receive a reward $y_t$.
4: **end for**
5: $X \leftarrow [x_1, ..., x_{t_1}]^\top$, $Y \leftarrow [y_t]_{t \in [t_1]}$.
6: Compute $\widehat{m}$ as (4).
7: **for** $t = t_2 + 1$ to $T$ **do**
8:     Playing OFUL algorithm [1] on $\widehat{m}$.
9: **end for**

---

$$\|\theta_\star - \Pi_m(\theta_\star)\|_2^2 \geq \varepsilon_0^2/2. \tag{11}$$

We now show that under the Assumption 17, after the exploring phase, the algorithm returns a true fixed-point subspace $\widehat{m} \ni \theta_\star$ with high probability.

**Theorem 18.** *Suppose the Assumptions 5, 6, 17 hold. Let* $t_2 = \Omega\left(\frac{\sigma^2 K_x^2 d_0 \log(dT)}{C_{\min}^2(\mathcal{X})\varepsilon_0^2}\right)$. *Then, Algorithm 2 returns* $\widehat{m} \ni \theta_\star$ *with probability at least* $1 - 1/T$, *and its regret is upper bounded as*

$$\mathbf{R}_T = O\left(\frac{R_{\max}\sigma^2 K_x^2 d_0 \log(dT)}{C_{\min}^2(\mathcal{X})\varepsilon_0^2} + \sigma d_0 \sqrt{T \log(K_x T)}\right). \tag{12}$$

That is, if the separating constant $\varepsilon_0$ is known in advance and $\varepsilon_0 \geq T^{-1/4}$, then we can achieve $O(d_0\sqrt{T}\log(K_x T))$ regret upper bound.

**Remark 19.** A weakness of Algorithm 2 is that without knowing that $\varepsilon_0 \geq T^{-1/4}$ is true a priori, there may be possible mis-specification error, which leads to linear regret if one applies the algorithm naively. On the other hand, Algorithm 1 can always achieve regret $O(T^{2/3})$ in the worst case. As such, the following question arises: *Does there exist an algorithm that, without the knowledge of $\varepsilon_0$, can achieve regret $O(\sqrt{T})$ whenever $\varepsilon_0 \geq T^{-1/4}$, but guarantees the worst-case regret of $O(T^{2/3})$?* Toward answering this question, we propose a simple method which has $O(\sqrt{T})$ regret whenever the separating constant is large, and enjoys a worst-case regret guarantee of $O(T^{3/4})$ (slightly worse than $O(T^{2/3})$). We refer the reader to Appendix C.2 for a detailed description of the algorithm, its regret bound and further discussion.

## 6  Experiment

To illustrate the performance of our algorithm, we conduct simulations where the entries of $\theta_\star$ satisfy three cases: sparsity, non-crossing partitions and non-nesting partitions. We refer readers to Appendix D.1 for a more formal description of non-crossing partitions, non-nesting partitions, and why the interval partition (i.e., the partition structure equivalent to sparsity) is a strict subset of both non-crossing and non-nesting partitions. Since sparsity is equivalent to a strict subset of non-crossing and non-nesting partitions, we compare our Algorithm 1 with the sparse-bandit ESTC algorithm proposed in [23] as a benchmark in all environments. The set of arms $\mathcal{X}$ is $\sqrt{d}\mathbb{S}^{d-1}$, $\sigma = 0.1$, and $d = 100$, $d_0 = 15$. The ground-truth sparse patterns, partitions and $\theta_\star$ are randomized before each simulation.

The regret of both algorithms is shown in Figure 2, which indicates that our algorithm performs competitively in the sparsity case and significantly outperforms the sparse-bandit algorithm in cases of non-crossing and non-nesting partitions. Due to space limitations, we refer the reader to Appendix E for a detailed description of the experiments, including how we applied the sparse-bandit algorithm in the cases of non-crossing and non-nesting partitions, and how we ran Algorithm 1 in the case of sparsity. Additionally, we explain how we exploited the particular structure of non-crossing and non-nesting partitions to enable efficient computation in Appendix E.

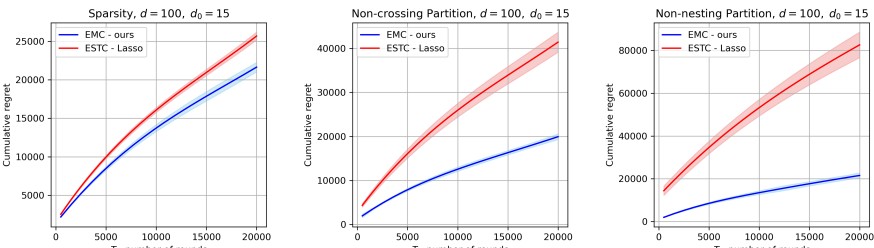

**Figure 2:** Regret of EMC (Algorithm 1) and of ESTC proposed in [23], in cases of sparsity, non-crossing partitions, and non-nesting partitions.

## 7  Conclusion and Future Work

In this paper, we study symmetric linear stochastic bandits in high dimensions, where the linear reward function is invariant with respect to some hidden subgroup $\mathcal{G} \leq \mathcal{S}_d$. We first prove that no algorithm can gain any advantage solely by knowing $\mathcal{G} \leq S_d$. Given this, we introduce a cardinality condition on the hidden subgroup $\mathcal{G}$, allowing the learner to overcome the curse of dimensionality. Under this condition, we propose novel model selection algorithms that achieve regrets of $\tilde{O}(d_0^{2/3} T^{2/3})$ and $\tilde{O}(d_0 \sqrt{T})$ with an additional assumption on the well-separated partition. For future work, we will explore convex relaxation techniques for efficient computation, leveraging specific structures of symmetries.

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

## Contents of Appendix

## Additional notations

For any subset $S \subseteq \mathbb{R}^d$, and matrix $W \in \mathbb{R}^{d \times d}$, we write $W(S) := \{Wx \mid x \in S\}$.

## A Impossibility Result of Learning with General Hidden Subgroups

To prove Proposition 1, we need to prove Proposition 20 and use Proposition 21.

**Proposition 20.** *For each $\Gamma \leq \mathcal{S}_d$ acting naturally on $[d]$, the orbit of $\Gamma$ on $[d]$ forms a unique partition of $[d]$. Moreover, for each partition $\rho \in \mathcal{P}_d$, there exists at least one $\Gamma \leq \mathcal{S}_d$ such that its orbit on $[d]$ under natural action is exactly $\rho$.*

*Proof.* The first claim is obvious by the property of orbit, that is, orbit consists of non-empty and disjoint subsets of $[d]$, whose union is $[d]$.

We prove the second claim. Let $\rho = \{\rho_i\}_{i \in I}$, where $I$ is the index of partition; note that $\rho_i$ is nonempty and mutually disjoint. Define a group as follows

$$\Gamma_i = \{f : \rho_i \to \rho_i \mid f \text{ is bijective}\}; \tag{13}$$

It is clear that $\Gamma_i$ is a group under function composition. Now, define the product group

$$\Gamma := \prod_{i \in I} \Gamma_i,$$

and the action $\psi : \Gamma \times [d] \to [d]$ such that

$$\psi\left((f_i)_{i \in I}, x\right) := f_j(x), \quad \text{for } \rho_j \ni x. \tag{14}$$

Therefore, it is clear that $(f_i)_{i \in I}$ is a bijection from $[d]$ onto itself, hence, $\Gamma$ is a subgroup of $\mathcal{S}_d$. $\quad\square$

Let $E = (e_i)_{i \in [d]}$ be the standard basis. Given that group $\mathcal{G}$ partition $[d]$ into $k$ disjoint orbits, $\mathcal{G}$ also partition $E$ into $k$ disjoint orbits corresponding to its action $\phi$ on $\mathbb{R}^d$, that is,

$$E = \bigcup_{i=1}^{k} E_i.$$

Let $V_i = \mathrm{Span}(E_i)$, then, one has the following.

**Proposition 21** ([10]'s Theorem 14)**.**

$$\mathrm{Fix}_{\mathcal{G}}(\mathbb{R}^d) = \bigoplus_{i=1}^{k} \mathrm{Fix}_{\mathcal{G}}(V_i). \tag{15}$$

We now state the proof of Proposition 1 as a corollary of Proposition 20 and Proposition 21.

**Proposition 1.** *There is a bijection* $\mathbf{H}$ *between* $\mathcal{P}_d$ *and* $\mathcal{F}_{\mathcal{S}_d}$.

*Proof.* **First**, we show that for each set partition of $[d]$, there exists a unique $H \in \mathcal{F}_{\mathcal{S}_d}(\mathbb{R}^d)$. This is straightforward due to the fact that, given a set partition $P$, the partition of basis $(E_i)_{i \in I}$ is unique by definition, so as $V_i = \mathrm{span}(E_i)$. As a result, let $H = \bigoplus_{i=1}^{k} \mathrm{Fix}_{\Gamma}(V_i)$, by Proposition 21, $H \in \mathcal{F}_{\mathcal{S}_d}(\mathbb{R}^d)$. As $(V_i)_{i \in I}$ is unique, $H$ is unique.

**Second**, we show that for each $H \in \mathcal{F}_{\mathcal{S}_d}(\mathbb{R}^d)$, there is a unique set partition/equivalent relation $P$. Denote $\overset{P}{\sim}$ as an equivalent relation under the set partition $P$. Suppose there are two different set partitions $P$, $Q$, there must be two set element $p$ $q$ such that $p \overset{P}{\sim} q$ under $P$, and $p \overset{Q}{\nsim} q$ under $Q$. Denote $H_P$, $H_Q$ as the subspaces defined by $P$ and $Q$ respectively. As $p \overset{Q}{\nsim} q$, there must exist a point $x \in H_Q$ such that $x_p \neq x_q$. However, $x_p = x_q$ for all $x \in H_P$. Hence, $H_P$ cannot be the same as $H_Q$. $\qquad\square$

**Proposition 3.** *Assume that the action set is the unit cube* $\mathcal{X} = \{x \in \mathbb{R}^d \mid \|x\|_\infty \leq 1\}$*, and* $f$ *is invariant w.r.t. action of subgroup* $\mathcal{G} \leq \mathcal{S}_d$*, such that* $\dim(\mathrm{Fix}_{\mathcal{G}}) = 2$*. Then, the regret of any bandit algorithm is lower bounded by* $\mathbf{R}_T = \Omega(d\sqrt{T})$.

*Proof.* Suppose there is a collection of model parameter $\Theta = \{-\varepsilon, \ \varepsilon\}^d$, for some $\varepsilon > 0$. For each $\theta \in \Theta$, it is straightforward that since $\theta$ has two classes of indices, by Proposition 1 and 2, there must be a subgroup $\mathcal{G} \leq \mathcal{S}_d$ such that $\theta \in \mathrm{Fix}_{\mathcal{G}}$ and $\dim(\mathrm{Fix}_{\mathcal{G}}) = 2$.

Now, $\Theta$ can be used as a family of problem instances for minimax lower bound of linear bandit. By Theorem 24.1 in [27], we have that with the choice of $\varepsilon = T^{-1/2}$, one has that

$$\mathbf{R}_T \geq \frac{\exp{-2}}{\sqrt{8}} d\sqrt{T}. \tag{16}$$

$\qquad\square$

# B    The Case of Hidden Subgroups with Subexponential Size

## B.1    Lower Bound for the Case of Hidden Subgroups with Subexponential Size

**Proposition 9** (**Regret lower bound**)**.** *There exist symmetric linear bandit instances in which Assumption 5, 6 hold with* $K_x = 8d_0$*, such that, any bandit algorithm must suffer regret* $\mathbf{R}_T = \Omega\left(\min\left(C_{\min}(\mathcal{X})^{-\frac{1}{3}} d_0^{\frac{2}{3}} T^{\frac{2}{3}}, \sqrt{dT}\right)\right)$.

*Proof sketch.* First, we show a change of variable between the sparsity and interval partition cases, including the parameter space and set of arms. Next, we demonstrate that mapping the problem instances in the lower bound of sparse bandits in [23, 24] satisfies the constraints in Assumption 6. After that, the proof follows immediately by carrying out step-by-step calculations similar to those in [23, 24].

First, let us define the sparse bandit instances. Let $\mathcal{Z}$ be the set of arm of sparse bandit problem, such that $\forall z \in \mathcal{Z}$, $\|z\|_\infty \leq 1$. Let $\varphi \in \mathbb{R}^d$ be a $d_0$-sparse parameters. Now, let us define the mapping that corresponds to change of variable between $\theta$ and $\varphi$ as follows: (1) For $i \in [d-1]$, $\varphi_i = \theta_i - \theta_{i+1}$, (2) $\varphi_d = \theta_d$. Therefore, one can write

$$\varphi = \underbrace{\begin{bmatrix} 1 & -1 & 0 & 0 & \cdots & 0 & 0 \\ 0 & 1 & -1 & 0 & \cdots & 0 & 0 \\ 0 & 0 & 1 & -1 & \cdots & 0 & 0 \\ \vdots & \vdots & \vdots & \vdots & \ddots & \vdots & \vdots \\ \vdots & \vdots & \vdots & \vdots & \ddots & \vdots & \vdots \\ 0 & 0 & 0 & 0 & \cdots & 1 & -1 \\ 0 & 0 & 0 & 0 & \cdots & 0 & 1 \end{bmatrix}}_{W \in \mathbb{R}^{d \times d}} \theta. \tag{17}$$

It is easy to verify that entries of $\theta$ has $k$ equivalent classes if and only if $\varphi$ is $k$ sparse. Now, suppose expected reward for a arm $x$ in interval-partition instance is equal to expected reward for an arm $z$ in sparsity instance, that is,

$$\langle x, \theta \rangle = \langle (W^\top)^{-1} x, W\theta \rangle = \langle z, \varphi \rangle. \tag{18}$$

Therefore, to guarantee the bijection between sparsity and interval partition instance, we define the set of arm $\mathcal{X} = W^\top(\mathcal{Z}) := \{ W^\top z \mid z \in \mathcal{Z} \}$.

We need to prove that $\mathcal{X}$ satisfies Assumption 6. Therefore, $x = W^\top z$, and $\mathcal{X} = W^\top(\mathcal{Z})$. Our job is to verify that, for all $z \in [-1, 1]^d$, the projection $\|\Pi_m(Vz)\|_2 \leq \sqrt{8d_0}$, for any $m$ such that $\dim(m) \leq 2d_0$. Let $S_1, ..., S_{\dim(m)}$ be the classes according to the interval partition corresponding to $m$, such that

$$\|\Pi_m(x)\|_2^2 = \sum_{k=1}^{\dim(m)} \left( \underbrace{\frac{1}{|S_k|} \sum_{i=(\sum_{j<k} |S_j|)+1}^{\sum_{j \leq k} |S_k|} x_i}_{\text{average of entries within class } S_k} \right)^2 |S_k|$$

$$= \sum_{k=1}^{\dim(m)} \frac{1}{|S_k|} \left( \sum_{i=(\sum_{j<k} |S_j|+1)}^{\sum_{j \leq k} |S_k|} x_i \right)^2. \tag{19}$$

Since $x = W^\top z$, we have that $x_i = -z_{i-1} + z_i$, $\forall i \in [d]$, with $z_0 = 0$ for convenience. Note that $z_i \in [-1, 1] \forall i \in [d]$. We have that

$$\|\Pi_m(W^\top z)\|_2^2 = \sum_{k=1}^{\dim(m)} \frac{1}{|S_k|} \left( \underbrace{\sum_{i=(\sum_{j<k} |S_j|+1)}^{\sum_{j \leq k} |S_j|} -z_{i-1} + z_i}_{\text{Telescoping sum}} \right)^2 \tag{20}$$

$$\leq \sum_{k=1}^{\dim(m)} \frac{4}{|S_k|} \leq 8d_0.$$

Therefore, we have that, for all $x \in \mathcal{X} = W^\top(\mathcal{Z})$, $\|\Pi_m(x)\|_2 \leq 2\sqrt{2d_0}$.

Moreover, we also have that $C_{\min}(\mathcal{X}) \geq C_{\min}(\mathcal{Z})$. Particularly, let us denote the exploratory distribution on $\mathcal{Z}$ as

$$\rho = \arg\max_{\omega \in \Delta(\mathcal{Z})} \lambda_{\min}\left(\mathbb{E}_{z \sim \omega}[zz^\top]\right), \quad P := \mathbb{E}_{z \sim \rho}[zz^\top], \quad C_{\min}(\mathcal{Z}) := \lambda_{\min}(P). \tag{21}$$

Define $\mu(x) := \rho((W^\top)^{-1}z)$, and let $U := \mathbb{E}_{x \sim \mu}[x^\top x]$, we have that $U = W^\top P W$. Since all eigenvalue of $W$ is 1, according to [30], we have that $\lambda_i(P) = \lambda_i(U)$ for all $i \in [d]$. Therefore, we have that $C_{\min}(\mathcal{X}) \geq C_{\min}(\mathcal{Z})$.

Then, using change of variable argument, the calculation of the lower bound is identical to that of [23, 24]. Therefore, we can conclude that all policies must suffer regret

$$\mathbf{R}_T = \Omega(C_{\min}(\mathcal{X})^{-\frac{1}{3}} d_0^{\frac{2}{3}} T^{\frac{2}{3}}, \sqrt{dT}).$$

$\square$

## B.2 High Probability Prediction Error of Model Selection

The proof of Proposition 14 uses the following concentration.

**Proposition 22** ([21]'s Theorem B.7). *For some subspace $S \subset \mathbb{R}^n$ of dimension $d_S$, with probability at least $1 - \delta$, one has that*

$$\left| \|\Pi_S(\boldsymbol{\eta})\|_2 - \sigma\sqrt{d_S} \right| \leq \sigma\sqrt{2\log\left(\frac{1}{\delta}\right)}. \tag{22}$$

**Proposition 14.** *Let $\boldsymbol{f}_\star = X\theta_\star$. For the choice of $\widehat{\boldsymbol{f}}_{\widehat{m}}$ as in Eqn. (3) & Eqn. (4), with probability at least $1 - \delta$, there exists a constant $C > 1$ such that*

$$\left\| \widehat{\boldsymbol{f}}_{\widehat{m}} - \boldsymbol{f}_\star \right\|_2^2 \leq C\sigma^2 \log\left( M\delta^{-1} \right). \tag{8}$$

*Proof.* Define $\mathcal{E}_1$ as the event such that for all $m \in \mathcal{M}$,

$$\|\Pi_{S_m}(\boldsymbol{\eta})\|_2 \leq \sigma\sqrt{d_m} + \sigma\sqrt{2\log\left(\frac{1}{\delta_0}\right)}. \tag{23}$$

By Proposition 22 and union bound, $\mathcal{E}_1$ occurs with probability $1 - M\delta_0$. For the rest of the proof, we assume $\mathcal{E}_1$ occurs .

Let us denote $\widehat{\boldsymbol{f}} := \widehat{\boldsymbol{f}}_{\widehat{m}}$. By the model selection procedure (4), one has that

$$\|Y - \widehat{\boldsymbol{f}}\|^2 \leq \|Y - \widehat{\boldsymbol{f}}_{m_\star}\|^2. \tag{24}$$

Also, as $Y = \boldsymbol{f}_\star + \boldsymbol{\eta}$, one has that, for some $K > 1$

$$\left\| \widehat{\boldsymbol{f}} - \boldsymbol{f}_\star \right\|_2^2 \leq \left\| \boldsymbol{f}_\star - \widehat{\boldsymbol{f}}_{m_\star} \right\|_2^2 + 2\left\langle \boldsymbol{\eta}, \boldsymbol{f}_\star - \widehat{\boldsymbol{f}}_{m_\star} \right\rangle + K\sigma^2 d_{\widehat{m}} - 2\left\langle \boldsymbol{\eta}, \boldsymbol{f}_\star - \widehat{\boldsymbol{f}} \right\rangle - K\sigma^2 d_{\widehat{m}}. \tag{25}$$

**First**, consider the term $\left\| \boldsymbol{f}_\star - \widehat{\boldsymbol{f}}_{m_\star} \right\|_2^2$. Note that, $\Pi_{S_{m_\star}}(\boldsymbol{f}_\star) = \boldsymbol{f}_\star$, as $\boldsymbol{f}_\star \in S_{m_\star}$. We have that,

$$\begin{aligned}
\left\| \boldsymbol{f}_\star - \widehat{\boldsymbol{f}}_{m_\star} \right\|_2^2 &= \left\| \boldsymbol{f}_\star - \Pi_{S_{m_\star}}(\boldsymbol{f}_\star + \boldsymbol{\eta}) \right\|_2^2 \\
&= \|\boldsymbol{f}_\star - \boldsymbol{f}_\star\|_2^2 - 2\left\langle \Pi_{S_{m_\star}}(\boldsymbol{\eta}), \boldsymbol{f}_\star - \boldsymbol{f}_\star \right\rangle + \left\| \Pi_{S_{m_\star}}(\boldsymbol{\eta}) \right\|_2^2 \\
&\leq \left( \sigma\sqrt{d_{m_\star}} + \sigma\sqrt{2\log\left(\frac{1}{\delta_0}\right)} \right)^2 \\
&\leq 2\sigma^2 d_0 + 4\sigma^2 \log\left(\frac{1}{\delta_0}\right).
\end{aligned} \tag{26}$$

**Second**, consider the term $\left\langle \boldsymbol{\eta}, \boldsymbol{f}_\star - \widehat{\boldsymbol{f}}_{m_\star} \right\rangle$. We have that,

$$
\begin{aligned}
\left\langle \boldsymbol{\eta}, \boldsymbol{f}_\star - \widehat{\boldsymbol{f}}_{m_\star} \right\rangle &= \left\langle \boldsymbol{\eta}, \boldsymbol{f}_\star - \Pi_{S_{m_\star}}(\boldsymbol{f}_\star + \boldsymbol{\eta}) \right\rangle \\
&= \left\langle \boldsymbol{\eta}, \boldsymbol{f}_\star - \boldsymbol{f}_\star \right\rangle - \left\| \Pi_{S_{m_\star}}(\boldsymbol{\eta}) \right\|_2^2 \le 0.
\end{aligned}
\tag{27}
$$

**Third**, we need to control the magnitude of the term $2 \left\langle \boldsymbol{\eta}, \widehat{\boldsymbol{f}} - \boldsymbol{f}_\star \right\rangle - K\sigma^2 d_{\widehat{m}}$. We show that this term can be bounded as

$$
2 \left\langle \boldsymbol{\eta}, \widehat{\boldsymbol{f}} - \boldsymbol{f}_\star \right\rangle - K\sigma^2 d_{\widehat{m}} \le a^{-1} \left\| \widehat{\boldsymbol{f}} - \boldsymbol{f}_\star \right\|_2^2 + O(\log(1/\delta_0)),
$$

with probability at least $1 - M\delta_0$, for any constant $a > 0$. Denote $\langle \boldsymbol{f}_\star \rangle$ be the line that is spanned by $\boldsymbol{f}_\star$. For each $m \in \mathcal{M}$, let us define the subspace $\bar{S}_m = S_m + \langle \boldsymbol{f}_\star \rangle$. Define $\tilde{S}_m \subset \bar{S}_m$ as the subspace that is orthogonal to $\langle \boldsymbol{f}_\star \rangle$, that is, one can write $\bar{S}_m = \tilde{S}_m \bigoplus \langle \boldsymbol{f}_\star \rangle$. By AM-GM inequality, for some $a > 0$, we have that

$$
\begin{aligned}
2 \left\langle \boldsymbol{\eta}, \widehat{\boldsymbol{f}} - \boldsymbol{f}_\star \right\rangle = 2 \left\langle \Pi_{\bar{S}_{\widehat{m}}}(\boldsymbol{\eta}), \widehat{\boldsymbol{f}} - \boldsymbol{f}_\star \right\rangle &= 2 \left\langle \sqrt{a} \cdot \Pi_{\bar{S}_{\widehat{m}}}(\boldsymbol{\eta}), \frac{1}{\sqrt{a}} \left( \widehat{\boldsymbol{f}} - \boldsymbol{f}_\star \right) \right\rangle \\
&\le a \left\| \Pi_{\bar{S}_{\widehat{m}}}(\boldsymbol{\eta}) \right\|_2^2 + a^{-1} \left\| \widehat{\boldsymbol{f}} - \boldsymbol{f}_\star \right\|_2^2 \\
&= a\sigma^2 V + a\sigma^2 U_{\widehat{m}} + a^{-1} \left\| \widehat{\boldsymbol{f}} - \boldsymbol{f}_\star \right\|_2^2,
\end{aligned}
\tag{28}
$$

where $V = \sigma^{-2} \left\| \Pi_{\langle \boldsymbol{f}_\star \rangle}(\boldsymbol{\eta}) \right\|_2^2$ and $U_{\widehat{m}} = \sigma^{-2} \left\| \Pi_{\tilde{S}_{\widehat{m}}}(\boldsymbol{\eta}) \right\|_2^2$. Note that, as $\dim(\langle \boldsymbol{f}_\star \rangle) = 1$, with probability at least $1 - \delta_0$, one has that

$$
V \le 2 + 4\log(\delta_0^{-1}).
\tag{29}
$$

Define the event $\mathcal{E}_2$ as the event where the above inequality holds.

Therefore, our final task is to control the quantity $aU_{\widehat{m}} - Kd_{\widehat{m}}$. Choose $a = (K+1)/2 > 1$, one has that

$$
aU_{\widehat{m}} - Kd_{\widehat{m}} = \frac{K+1}{2} \left( U_{\widehat{m}} - \frac{2K}{K+1} d_{\widehat{m}} \right) \le \frac{K+1}{2} \max_{m \in \mathcal{M}} \left( U_m - \frac{2K}{K+1} d_m \right).
$$

Now, directly control the magnitude of $U_{\widehat{m}} - \frac{2K}{K+1} d_{\widehat{m}}$ is difficult, as $\widehat{m}$ depends on $\boldsymbol{\eta}$, and their distribution might be complicated. Instead, we will control the maximum of the above quantity $U_m - \frac{2K}{K+1} d_m$ over all $m \in \mathcal{M}$. Since the dimension of $\tilde{S}_m$ is at most $d_m$, similar as the event $\mathcal{E}_1$, we define the event $\mathcal{E}_3$ as for all $m \in \mathcal{M}$, we have that

$$
\left\| \Pi_{\tilde{S}_m}(\boldsymbol{\eta}) \right\|_2 \le \sigma\sqrt{d_m} + \sigma\sqrt{2\log\left(\frac{1}{\delta_0}\right)}.
\tag{30}
$$

By Proposition 22 and the union bound, $\mathcal{E}_3$ occurs with probability at least $1 - M\delta_0$. We assume that $\mathcal{E}_3$ occurs, then for all $m \in \mathcal{M}$,

$$
\begin{aligned}
U_m &\le \left( \sqrt{d_m} + \sqrt{2\log\left(\frac{1}{\delta_0}\right)} \right)^2 \\
&\le \frac{2K}{K+1} d_m + \frac{4K}{K-1} \log\left(\frac{1}{\delta_0}\right).
\end{aligned}
\tag{31}
$$

Therefore, one has that for all $m \in \mathcal{M}$,

$$\frac{K+1}{2}\left(U_{\widehat{m}} - \frac{2K}{K+1}d_{\widehat{m}}\right) \leq \frac{K+1}{2}\max_{m \in \mathcal{M}}\left(U_m - \frac{2K}{K+1}d_m\right)$$
$$\leq \frac{2K(K+1)}{K-1}\log\left(\frac{1}{\delta_0}\right). \tag{32}$$

**Putting things together,** let $\delta = (2M+1)\delta_0$, we assure that event $\mathcal{E}_1 \cap \mathcal{E}_2 \cap \mathcal{E}_3$ occurs with probability at least $1 - \delta$. Combining (26), (27), (29), (32), with (25), one has that

$$\frac{K-1}{K+1}\left\|\widehat{\boldsymbol{f}} - \boldsymbol{f}_\star\right\|_2^2 \leq 2\sigma^2 d_0 + 4\sigma^2 \log\left(\frac{2M+1}{\delta}\right) + K\sigma^2 d_0$$
$$+ \sigma^2(K+1) + 2\sigma^2(K+1)\log\left(\frac{2M+1}{\delta}\right) + \sigma^2\frac{2K(K+1)}{K-1}\log\left(\frac{2M+1}{\delta}\right). \tag{33}$$

Note that, the dominating term in the above equation is $\log\left(\frac{M}{\delta}\right)$, we have that, there is some constant $C > 0$.

$$\left\|\widehat{\boldsymbol{f}} - \boldsymbol{f}_\star\right\|_2^2 \leq C\sigma^2 \log\left(\frac{M}{\delta}\right). \tag{34}$$

$\square$

**Proposition 16.** *Let $X = [x_t]_{t \in [n]}$, where $x_t$'s are is i.i.d. drawn from $\nu$, and let $n = \Omega\left(C_{\min}^{-2}(\mathcal{X})K_x^2\left(\log(2\overline{M}d_0\delta^{-1})\right)\right)$. Then, with probability at least $1 - \delta$, and for any $\theta_1, \theta_2$ in subspaces of $\mathcal{M}$, one has that*

$$\|\theta_1 - \theta_2\|_2^2 \leq 2C_{\min}^{-1}(\mathcal{X})n^{-1}\|X(\theta_1 - \theta_2)\|_2^2. \tag{10}$$

*Proof.* Our proof strategy is inspired by [8].

First, we compute the subgaussian norm for the normalised distribution along fews directions $\nu$. For any $x \in \mathcal{X}$, we have that

$$\max_{\substack{\|v\|_2=1, \\ v \in V^{1/2}(\overline{m}),\ \overline{m} \in \overline{\mathcal{M}}}} \left\langle V^{-1/2}x, v\right\rangle^2 = \max_{\substack{\|V^{1/2}\omega\|_2=1, \\ \omega \in \overline{m},\ \overline{m} \in \overline{\mathcal{M}}}} \left\langle V^{-1/2}x, V^{1/2}\omega\right\rangle^2$$
$$= \max_{\substack{\|V^{1/2}\omega\|_2=1, \\ \omega \in \overline{m},\ \overline{m} \in \overline{\mathcal{M}}}} \langle x, \omega\rangle^2 \tag{35}$$
$$\leq \max_{\overline{m} \in \overline{\mathcal{M}}} \|\Pi_{\overline{m}}(x)\|_2^2\, C_{\min}^{-1}(\mathcal{X}) \leq \frac{K_x}{C_{\min}(\mathcal{X})}.$$

This means that, the distribution $\left\langle V^{-1/2}x, v\right\rangle^2$ has bounded support $[0, \frac{K_x}{C_{\min}}]$ for any above direction of $v$.

Next, for any $\theta \in \overline{m}$, denote $\theta_V := V^{1/2}\theta$, note that the direction $v$ corresponding to $\theta_V$ satisfies (35). Using Hoeffding's inequality for bounded random variable, there is an absolute constant $c_1 > 0$ such that

$$\Pr\left\{\left|\frac{1}{n}\sum_{i=1}^n \left\langle V^{-1/2}x_i, \theta_V\right\rangle^2 - \|\theta_V\|_2^2\right| \geq \epsilon\|\theta_V\|_2^2\right\} \leq 2\exp\left(-\frac{nc_1 C_{\min}^2(\mathcal{X})\epsilon^2}{K_x^2}\right)$$
$$\iff \Pr\left\{\left|\frac{1}{n}\left\|XV^{-1/2}\theta_V\right\|_2^2 - \|\theta_V\|_2^2\right| \geq \epsilon\|\theta_V\|_2^2\right\} \leq 2\exp\left(-\frac{nc_1 C_{\min}^2(\mathcal{X})\epsilon^2}{K_x^2}\right). \tag{36}$$

Let $Z := \frac{1}{\sqrt{n}}XV^{-1/2}$. From Lemma 5.1 in [5], we know that if the above inequality holds, then

$$(1 - \delta_{\overline{M}}(Z)\|\theta_V\|_2) \leq \|Z\theta_V\|_2 \leq (1 + \delta_{\overline{M}}(Z)\|\theta_V\|_2), \tag{37}$$

holds with probability more than

$$1 - 2 \left( \frac{12}{\delta_{\overline{M}}(Z)} \right)^{2d_0} \exp \left( -\frac{c_1 C_{\min}^2(\mathcal{X}) n \delta_{\overline{M}}^2(Z)}{K_x^2} \right),$$

for any $\theta_V \in V^{1/2}(\overline{m})$ in a subspace $\overline{m} \in \overline{M}$. Take union bound for $\overline{M}$ subspaces, and let

$$\delta = 2\overline{M} \left( \frac{2}{\delta_{\overline{M}}(Z)} \right)^{2d_0} \exp \left( -\frac{c_1 C_{\min}^2(\mathcal{X}) n \delta_{\overline{M}}^2(Z)}{K_x^2} \right),$$

$$\iff n = O \left( \frac{K_x^2}{\delta_{\overline{M}}^2(Z) C_{\min}^2(\mathcal{X})} \left( \log(2\overline{M}) + d_0 \log \left( \frac{1}{\delta_{\overline{M}}(Z)} \right) + \log(\delta^{-1}) \right) \right).$$

Let $\delta_{\overline{M}}^2(Z) = 1/2$. Then, Given $n = O \left( \frac{K_x^2}{C_{\min}^2(\mathcal{X})} \left( \log(2\overline{M}) + d_0 + \log(\delta^{-1}) \right) \right)$, for probability at least $1 - \delta$, we have that

$$\frac{1}{2} \left\| V^{1/2}\theta \right\|_2^2 \quad \leq \frac{1}{n} \left\| XV^{-1/2}V^{1/2}\theta \right\|_2^2,$$

$$\implies \frac{1}{2} C_{\min}(\mathcal{X}) \|\theta\|_2^2 \quad \leq \frac{1}{n} \|X\theta\|_2^2.$$

Let $\theta := \theta_1 - \theta_2$, for any $\theta_1 \in m, \theta_2 \in m'$, for any $m, m' \in \mathcal{M}$. We conclude the proof. $\qquad\square$

## B.3   Regret Upper bound

**Lemma 12.** *Suppose the Assumptions 5, 6 hold. For $t_1 = \Omega(K_x^2 d_0 C_{\min}^{-2}(\mathcal{X}) \log(d/\delta))$, with probability at least $1 - \delta$, one has the estimate*

$$\left\| \theta_\star - \widehat{\theta}_{t_1} \right\|_2 = O \left( \sqrt{\frac{\sigma^2 d_0 \log(d/\delta)}{C_{\min}(\mathcal{X}) t_1}} \right). \tag{7}$$

*Proof.* First, let consider Proposition 16, and recall that $\log(M)$ and $\log(\overline{M})$ are both $O(d_0 \log(d))$. Therefore, for the choice of $t_1 = \Omega(K_x^2 d_0 C_{\min}^{-2}(\mathcal{X}) \log(d/\delta))$, with probability at least $1 - \delta/2$, one has that

$$C_{\min}(\mathcal{X}) \left\| \theta_\star - \widehat{\theta}_{t_1} \right\|_2^2 \leq \frac{2}{t_1} \left\| X(\theta_\star - \widehat{\theta}_{t_1}) \right\|_2^2 \tag{38}$$

Second, by Proposition 14, one has that with probability at least $1 - \delta/2$.

$$\left\| X(\theta_\star - \widehat{\theta}_{t_1}) \right\|_2^2 \leq c_1 \sigma^2 d_0 \log \left( \frac{d}{\delta} \right), \tag{39}$$

for some constant $c_1 > 0$.

Therefore, putting thing together, we have that with probability at least $1 - \delta$, one has that

$$\left\| \theta_\star - \widehat{\theta}_{t_1} \right\|_2 \leq c_2 \sqrt{\frac{\sigma^2 d_0 \log \left( \frac{d}{\delta} \right)}{C_{\min}(\mathcal{X}) t_1}}, \tag{40}$$

for some constant $c_2 > 0$. $\qquad\square$

**Theorem 10** (**Regret upper bound**). *Suppose the Assumptions 5, 6 hold. With the choice of $t_1 = R_{\max}^{-\frac{2}{3}} \sigma^{\frac{2}{3}} C_{\min}^{-\frac{1}{3}}(\mathcal{X}) K_x^{\frac{1}{3}} d_0^{\frac{1}{3}} T^{\frac{2}{3}} (\log(dT))^{\frac{1}{3}}$, then the regret of Algorithm 1 is upper bounded as*

$$\mathbf{R}_T = O \left( R_{\max}^{\frac{1}{3}} \sigma^{\frac{2}{3}} C_{\min}^{-\frac{1}{3}}(\mathcal{X}) K_x^{\frac{1}{3}} d_0^{\frac{1}{3}} T^{\frac{2}{3}} (\log(dT))^{\frac{1}{3}} \right) \tag{6}$$

*Proof.* Let define the pesudo regret as $\widehat{\mathbf{R}}_T = \sum_{t=1}^T \langle x_\star - x_t, \theta_\star \rangle$. Denote $\overline{m} \in \overline{M}$ be the subspace contains $\theta_\star - \widehat{\theta}_{t_1}$. We start by simple regret decomposition as follows.

$$\widehat{\mathbf{R}}_T = \sum_{t=1}^{T} \langle \theta_\star, x_\star - x_t \rangle = \sum_{t=1}^{t_1} \langle \theta_\star, x_\star - x_t \rangle + \sum_{t=t_1+1}^{T} \langle \theta_\star, x_\star - x_t \rangle$$

$$\le R_{\max} t_1 + \sum_{t=t_1+1}^{T} \left\langle \theta_\star - \widehat{\theta}_{t_1}, x_\star - x_t \right\rangle + \sum_{t=t_1+1}^{T} \underbrace{\left\langle \widehat{\theta}_{t_1}, x_\star - x_t \right\rangle}_{\le 0}$$

$$\le R_{\max} t_1 + \sum_{t=t_1+1}^{T} \left\langle \theta_\star - \widehat{\theta}_{t_1}, x_\star - x_t \right\rangle$$

$$= R_{\max} t_1 + \sum_{t=t_1+1}^{T} \left\langle \theta_\star - \widehat{\theta}_{t_1}, \Pi_{\overline{m}}(x_\star - x_t) \right\rangle \qquad \left[ \text{As } \theta_\star - \widehat{\theta}_{t_1} \in \overline{m}, \right]$$

$$\le R_{\max} t_1 + \sum_{t=t_1+1}^{T} \left\| \theta_\star - \widehat{\theta}_{t_1} \right\|_2 \left\| \Pi_{\overline{m}}(x_\star - x_t) \right\|_2$$

$$\le R_{\max} t_1 + \sum_{t=t_1+1}^{T} 2\sqrt{K_x} \left\| \theta_\star - \widehat{\theta}_{t_1} \right\|_2 ; \qquad \left[ \text{Since } \| \Pi_{\overline{m}}(x_\star - x_t) \|_2 \le 2\sqrt{K_x} \right].$$

$$\tag{41}$$

Now, we invoke Lemma 12. Let $\mathcal{E}$ is the event in that the exploration error is bounded as in Lemma 12, then there is an absolute constant $c_1 > 0$ such that,

$$\mathbf{R}_T \le R_{\max} t_1 + \mathbb{E}\left[ \sum_{t=t_1+1}^{T} 2\sqrt{K_x} \left\| \theta_\star - \widehat{\theta}_{t_1} \right\|_2 \Big| \mathcal{E} \right] + T \Pr(\mathcal{E}) R_{\max}$$

$$\tag{42}$$

$$\le R_{\max} t_1 + c_1 \sqrt{K_x} \sqrt{\frac{\sigma^2 d_0 \log(d/\delta)}{C_{\min}(\mathcal{X}) t_1}} T + T\delta R_{\max}.$$

Let $\delta = 1/T$, and $t_1 = R_{\max}^{-\frac{2}{3}} \sigma^{\frac{2}{3}} C_{\min}^{-\frac{1}{3}}(\mathcal{X}) K_x^{\frac{1}{3}} d_0^{\frac{1}{3}} T^{\frac{2}{3}} (\log(dT))^{\frac{1}{3}}$, one has that

$$\mathbf{R}_T = O\left( R_{\max}^{\frac{1}{3}} \sigma^{\frac{2}{3}} C_{\min}^{-\frac{1}{3}}(\mathcal{X}) K_x^{\frac{1}{3}} d_0^{\frac{1}{3}} T^{\frac{2}{3}} (\log(dT))^{\frac{1}{3}} \right)$$

$$\tag{43}$$

$\square$

# C  Improved Regret Bound

## C.1  Improve Regret Bound with Well-Separated Partitions

**Theorem 18.** *Suppose the Assumptions 5, 6, 17 hold. Let* $t_2 = \Omega\left( \frac{\sigma^2 K_x^2 d_0 \log(dT)}{C_{\min}^2(\mathcal{X}) \varepsilon_0^2} \right)$. *Then, Algorithm 2 returns* $\widehat{m} \ni \theta_\star$ *with probability at least* $1 - 1/T$, *and its regret is upper bounded as*

$$\mathbf{R}_T = O\left( \frac{R_{\max} \sigma^2 K_x^2 d_0 \log(dT)}{C_{\min}^2(\mathcal{X}) \varepsilon_0^2} + \sigma d_0 \sqrt{T \log(K_x T)} \right).$$

$$\tag{12}$$

*Proof.* Define the event $\mathcal{E}$ as

$$\mathcal{E} = \left\{ \| Y - \Pi_{S_{\widehat{m}}}(Y) \|_2^2 \le \| Y - \Pi_{S_m}(Y) \|_2^2 \mid \theta_\star \in \widehat{m}, \ \theta_\star \notin m \right\},$$

$$\tag{44}$$

The event $\mathcal{E}$ is equivalent as

$$\|Y - \Pi_{S_{\widehat{m}}}(Y)\|_2^2 \leq \|Y - \Pi_{S_m}(Y)\|_2^2$$

$$\Longleftrightarrow \quad \left\|\boldsymbol{f}_\star + \boldsymbol{\eta} - \widehat{\boldsymbol{f}}_{\widehat{m}}\right\|_2^2 \leq \left\|\boldsymbol{f}_\star + \boldsymbol{\eta} - \widehat{\boldsymbol{f}}_m\right\|_2^2 \tag{45}$$

$$\Longleftrightarrow \quad \left\|\boldsymbol{f}_\star - \widehat{\boldsymbol{f}}_{\widehat{m}}\right\|_2^2 + 2\left\langle \boldsymbol{\eta}, \boldsymbol{f}_\star - \widehat{\boldsymbol{f}}_{\widehat{m}}\right\rangle \leq \left\|\boldsymbol{f}_\star - \widehat{\boldsymbol{f}}_m\right\|_2^2 + 2\left\langle \boldsymbol{\eta}, \boldsymbol{f}_\star - \widehat{\boldsymbol{f}}_m\right\rangle.$$

**First**, we upper the LHS of (45) with high probability.

$$\|\boldsymbol{f}_\star - \boldsymbol{f}_\star - \Pi_{S_{\widehat{m}}}(\boldsymbol{\eta})\|_2^2 + 2\left\langle \boldsymbol{\eta}, -\Pi_{S_{\widehat{m}}}(\boldsymbol{\eta})\right\rangle = \|\Pi_{S_{\widehat{m}}}(\boldsymbol{\eta})\|_2^2 - 2\|\Pi_{S_{\widehat{m}}}(\boldsymbol{\eta})\|_2^2 \leq 0 \quad [\text{as } \theta_\star \in \widehat{m}] \tag{46}$$

with probability at least $1 - \delta$. The above inequality uses the union bound for all $m \in \mathcal{M}$.

**Second**, we lower bound the RHS of (45), $\left\|\boldsymbol{f}_\star - \widehat{\boldsymbol{f}}_m\right\|_2^2 + 2\left\langle \boldsymbol{\eta}, \boldsymbol{f}_\star - \widehat{\boldsymbol{f}}_m\right\rangle$, with high probability. Let $\bar{S}_m = S_m \bigoplus \langle \boldsymbol{f}_\star \rangle$, then $\left\langle \boldsymbol{\eta}, \boldsymbol{f}_\star - \widehat{\boldsymbol{f}}_m\right\rangle = \left\langle \Pi_{\bar{S}_m}(\boldsymbol{\eta}), \boldsymbol{f}_\star - \widehat{\boldsymbol{f}}_m\right\rangle$. Therefore, we have that, with probability at least $1 - \delta/3$.

$$\left\|\boldsymbol{f}_\star - \widehat{\boldsymbol{f}}_m\right\|_2^2 + 2\left\langle \Pi_{\bar{S}_m}(\boldsymbol{\eta}), \boldsymbol{f}_\star - \widehat{\boldsymbol{f}}_m\right\rangle \geq \left\|\boldsymbol{f}_\star - \widehat{\boldsymbol{f}}_m\right\|_2^2 - 2\|\Pi_{\bar{S}_m}(\boldsymbol{\eta})\|_2^2 - \frac{1}{2}\left\|\boldsymbol{f}_\star - \widehat{\boldsymbol{f}}_m\right\|_2^2$$

$$\geq \frac{1}{2}\left\|\boldsymbol{f}_\star - \widehat{\boldsymbol{f}}_m\right\|_2^2 - 4\left(\sigma^2(d_0 + 1) + \sigma^2(\log(3M\delta^{-1}))\right)$$

$$\geq \frac{1}{2}\left\|\boldsymbol{f}_\star - \widehat{\boldsymbol{f}}_m\right\|_2^2 - 5\left(\sigma^2 d_0 + \sigma^2(\log(3M\delta^{-1}))\right), \tag{47}$$

as $d_m \leq d_0$.

Also, we have that, with probability at least $1 - \delta/3$

$$\left\|\boldsymbol{f}_\star - \widehat{\boldsymbol{f}}_m\right\|_2^2 = \|\boldsymbol{f}_\star - \Pi_{S_m}(\boldsymbol{f}_\star) - \Pi_{S_m}(\boldsymbol{\eta})\|_2^2$$

$$\geq \|\boldsymbol{f}_\star - \Pi_{S_m}(\boldsymbol{f}_\star)\|_2^2$$

Where the inequalities holds because $\Pi_{S_m}(\boldsymbol{f}_\star)$ is the projection of $\boldsymbol{f}_\star$ to $S_m$. Therefore, we have that

$$\left\|\boldsymbol{f}_\star - \widehat{\boldsymbol{f}}_m\right\|_2^2 + 2\left\langle \boldsymbol{\eta}, \boldsymbol{f}_\star - \widehat{\boldsymbol{f}}_m\right\rangle \geq \frac{1}{2}\|\boldsymbol{f}_\star - \Pi_{S_m}(\boldsymbol{f}_\star)\|_2^2 - 5\left(\sigma^2 d_0 + \sigma^2(\log(M\delta^{-1}))\right). \tag{48}$$

Now, by choosing $t_2 = \Omega\left(C_{\min}^{-2}(\mathcal{X})K_x^2\log(M\delta^{-1})\right)$, by Proposition 16, we have that with probability at least $1 - \delta/3$,

$$\|\boldsymbol{f}_\star - \Pi_{S_m}(\boldsymbol{f}_\star)\|_2^2 = \|X(\theta_\star - \Pi_{S_m}(\theta_\star))\|_2^2 \geq \frac{t_2}{2C_{\min}(\mathcal{X})}\|\theta_\star - \Pi_{S_m}(\theta_\star)\|_2^2 \geq \frac{t_2\varepsilon_0^2}{4C_{\min}(\mathcal{X})} \tag{49}$$

Therefore, the RHS of (45) is lower bounded as follows

$$\left\|\boldsymbol{f}_\star - \widehat{\boldsymbol{f}}_m\right\|_2^2 + 2\left\langle \boldsymbol{\eta}, \boldsymbol{f}_\star - \widehat{\boldsymbol{f}}_m\right\rangle \geq \frac{t_2\varepsilon_0^2}{8C_{\min}(\mathcal{X})} - 5\left(\sigma^2 d_0 + \sigma^2(\log(3M\delta^{-1}))\right). \tag{50}$$

Therefore, the sufficient condition for event $\mathcal{E}$ holds with probability at least $1 - \delta$ is that

$$\frac{t_2\varepsilon_0^2}{8C_{\min}(\mathcal{X})} - 5\left(\sigma^2 d_0 + \sigma^2(\log(M\delta^{-1}))\right) \geq 0; \tag{51}$$

Note that as $M = O(d_0\log(d))$, for $\mathcal{E}$ holds with probability at least $1 - \delta$, it suffice to choose

$$t_2 = \Omega\left(\frac{\sigma^2 K_x^2 d_0\log(d\delta^{-1})}{C_{\min}^2(\mathcal{X})\varepsilon_0^2}\right). \tag{52}$$

The regret upper bound is an immediate consequence of the fact that Algorithm 2 return the true subspace $\hat{m} \ni \theta_\star$, combined with the regret bound of OFUL in the exploitation phase and $\delta = 1/T$. □

## C.2 Adapting to Separating Constant $\varepsilon_0$

As stated in Theorem 18, if one knows in advance that the separating constant $\varepsilon_0 \geq T^{-1/4}$, then using Algorithm 2 leads to $\sqrt{T}$ regret. The reason is that the learner can learn the true subspace $m_\star$ after $\sqrt{T}$ steps with no mis-specification errors. However, without knowing $\varepsilon_0 \geq T^{-1/4}$ a priori, one cannot guarantee to recover the true subspace. Hence, naively using Algorithm 2, which is not aware of potential mis-specification errors, leads to linear regret. On the other hand, using Algorithm 1 can achieve regret $T^{2/3}$ in the worst case (without the knowledge of $\varepsilon_0 \geq T^{-1/4}$). A question arises: *does there exist an algorithm that, without the knowledge of $\varepsilon_0$, can achieve regret $\sqrt{T}$ whenever $\varepsilon_0 \geq T^{-1/4}$, but guarantee the worst-case regret as $T^{2/3}$?*

We note that the role of $\varepsilon_0$ is similar to the minimum signal in sparsity, and it is somewhat surprising that the question of adapting to unknown minimum signal has not been resolved in the literature of sparse linear bandits. Towards answering the question, we propose a simple method using adaptation to misspecified error in linear bandit [19], which has a $\sqrt{T}$ regret whenever the separating constant is large, and enjoys a worst-case regret guarantee of slightly worse $T^{3/4}$ regret.

The algorithm described in 4 is a direct application of the algorithm proposed in [19], designed for adapting to misspecification errors in linear bandits. Particularly, the algorithm in [19] can adapt to unknown misspecification errors and achieve a regret bound of $\tilde{O}(d_0\sqrt{T} + \epsilon_{\text{mis}}T)$, where $\epsilon_{\text{mis}}$ is the misspecification error. At a high level, our algorithm exploring $\sqrt{T}$ rounds using exploratory distribution, which ensures that the misspecification error $\epsilon_{\text{mis}}$ of the chosen subspace $\hat{m}$ is at most $T^{-1/4}$. Therefore, we can run multiple linear bandit algorithms using different levels of misspecification error. Particularly, we use a collection of $K = \lfloor \log(T) \rfloor$ base algorithms, where a base algorithm $k \in [K]$ is a linear bandit algorithm with misspecified level $\varepsilon_k = 2^{-k}$. Note that the base algorithm $K$ has the same order of regret $\sqrt{T}$ as a well-specified model. Therefore, in exploitation phase, one can guarantee that in the case of well-separated partitions where $\epsilon_{\text{mis}} = 0$, the algorithm can achieve a regret of $d_0\sqrt{T}$, while in the general case, the regret caused by misspecification error is at most $T^{3/4}\sqrt{d_0}$.

---

**Algorithm 3** Adaptive algorithm

---
1: Input $T$, $\nu$, $t_3$.
2: **for** $t = 1, \cdots, t_3$ **do**
3:   Independently pull arm $x_t$ according to $\nu$ and receive a reward $y_t$.
4: **end for**
5: $X \leftarrow [x_1, ..., x_{t_1}]^\top, Y \leftarrow [y_t]_{t \in [t_1]}$.
6: Compute $\hat{m}$ as (4).
7: Let $K = \lfloor \log(T^{1/4}) \rfloor, \mathcal{E} = \{\varepsilon_k := 2^{-k}, \ k \in [K]\}$.
8: **for** $t = t_2 + 1$ to $T$ **do**
9:   Corralling $K$ base misspecified linear bandit algorithms `SquareCB.Lin+`$(\varepsilon_k)$ [19] on $\hat{m}$.
10: **end for**

---

**Corollary 23.** *Suppose the Assumptions 5, 6 hold. Then, there exists an algorithm which achieves regret bound as follows:*

   *(i) [**Well-separated partitions**] If $\varepsilon_0 \geq T^{-1/4}$, then $\mathbf{R}_T = \tilde{O}(d_0\sqrt{T})$.*

   *(ii) [**Non-well-separated partitions**] If $\varepsilon_0 < T^{-1/4}$, then $\mathbf{R}_T = \tilde{O}(d_0\sqrt{T} + T^{\frac{3}{4}}\sqrt{d_0})$.*

*Proof sketch.* Denote $\epsilon_{\text{mis}} = \|\theta_\star - \Pi_{\hat{m}}(\theta_\star)\|_2$ as the misspecification error. With

$$t_3 = \Omega\left(\sigma^2 d_0 \log(d\delta^{-1})\sqrt{T}\right),$$

We can guarantee that, with probability at least $1 - \delta$, if:

(i) If $\varepsilon_0 > T^{-1/4}$, then by Theorem 18, $\theta_\star \in \widehat{m}$, that is, $\epsilon_{\text{mis}} = 0$;

(ii) If $\varepsilon_0 > T^{-1/4}$, then by Lemma 12, we can bound $\epsilon_{\text{mis}} \leq T^{-1/4}$.

The regret of adaptive algorithm in [19] is of the form $\tilde{O}(d_0\sqrt{T} + \epsilon_{\text{mis}}T\sqrt{d_0})$. Let $\delta = 1/T$. Consider case (i) where $\epsilon_{\text{mis}} = 0$, we have $\mathbf{R}_T = \tilde{O}(d_0\sqrt{T})$. Consider case (ii), where $\epsilon_{\text{mis}} = T^{-1/4}$, we have $\mathbf{R}_T = \tilde{O}(d_0\sqrt{T} + T^{3/4}\sqrt{d_0})$. $\square$

The result in Corollary 23 is still sub-optimal in the worst case, as it can only achieve $O(T^{3/4})$ regret bound instead of $O(T^{2/3})$. We conjecture that new techniques are required to achieve order-optimal regret in both cases, and will continue to investigate this question in future works.

## D On Collections of Partitions with Subexponential-Size

### D.1 Important Classes of Partitions with Subexponential-Size

In this section, we discuss several important classes of partitions which satisfy Assumption 5.

Pattern-avoidance partitions is arguable the most important class of studied partition [33], in which, non-crossing partition is one the most studied.

**Definition 24** (**Non-Crossing Partition**). Let $[d]$ admits a cyclic order as $1 < 2 < ... < d$, and $d < 1$. A non-crossing partition of $[d]$ is a partition such that for if $i$, $j$ in one block and $p$, $q$ in one block, then they are not arranged in the order $i < p < j < q$.

Similarly, we denote $\mathcal{NC}_d, \mathcal{NC}_{d,k}, \mathcal{NC}_{d,\leq k}$ as the set of all non-crossing partition of $[d]$, the set of all partition of $[d]$ with $k$ classes, and the set of all partition of $[d]$ with at most $k$ classes. We have the following fact ([33]' section 3.2).

$$|\mathcal{NC}_d| = \frac{1}{d+1}\binom{2d}{d}, \quad |\mathcal{NC}_{d,k}| = \frac{1}{d}\binom{d}{k}\binom{d}{k-1}, \tag{53}$$

which is the Catalan number and the Narayana number. Note that,

$$\frac{1}{d}\binom{d}{k}\binom{d}{k-1} \leq \frac{1}{d}\left(\frac{ed}{k}\right)^k\left(\frac{ed}{k-1}\right)^{k-1} \leq \frac{1}{d}\left(\frac{ed}{k}\right)^{2k}. \tag{54}$$

Therefore, non-crossing partition satisfied the cardinality restriction as Assumption 5, that is, $\mathcal{NC}_{d,\leq d_0} \subset \mathcal{Q}_{d,\leq d_0}$.

Another important class of pattern-avoidence partitions is nonnesting partition [14].

**Definition 25** (**Non-Nesting Partition**). Let $[d]$ admits a cyclic order as $1 < 2 < ... < d$, and $d < 1$. A non-crossing partition of $[d]$ is a partition such that for if $i$, $j$ in one block and $p$, $q$ in one block, then they are not arranged in the order $i < p < q < j$.

Similarly, we denote $\mathcal{NN}_d, \mathcal{NN}_{d,k}, \mathcal{NN}_{d,\leq k}$ as the set of all non-crossing partition of $[d]$, the set of all partition of $[d]$ with $k$ classes, and the set of all partition of $[d]$ with at most $k$ classes. There is bijections between class of $\mathcal{NN}_d$ and $\mathcal{NC}_d$, non-nesting partitions also satisfy the sub-exponential constraint (Assumption 5).

One special case of both non-crossing partitions and non-nesting partitions is interval partition, which has *identical structure as sparsity*.

**Definition 26** (**Interval Partition**). A set partition of $[d]$ is an interval partition or partition of interval if its parts are interval.

We denote $\mathcal{I}_d$ as the collection of all interval partition of $d$, we have that $\mathcal{I}_d \subset \mathcal{NC}_d \subset \mathcal{P}_d$, and $\mathcal{I}_d \subset \mathcal{NN}_d \subset \mathcal{P}_d$.

**Remark 27.** $\mathcal{I}_d$ admits a Boolean lattice of order $2^{d-1}$, making it equivalent to the sparsity structure in $d-1$ dimensions. Specifically, consider the set of entries of parameters $\varphi \in \mathbb{R}^d$ with a linear order, that is, $\varphi_1 < \varphi_2 < \cdots < \varphi_d$. Then define the variable $\theta \in \mathbb{R}^{d-1}$ such that $\theta_i = (\varphi_{i+1} - \varphi_i)$. Each interval partition on the entries of $\varphi$ will determine a unique sparse pattern of $\theta$. In other words, symmetric linear bandit is strictly harder than sparse bandit and inherits all the computational complexity challenges of sparse linear bandit, including the *NP-hardness* of computational complexity.

Inspired by the literature on sparse linear regression, where one can relax solving exact sparse linear regression by using norm-1 minimization, also known as LASSO methods, we ask whether there is a convex relaxation for the case of non-crossing partitions or pattern-avoidance partitions in general.

## D.2 Practical Examples of Partitions with Subexponential-Size

**General hidden symmetries.** Examples of hidden symmetry in reinforcement learning tasks can be found in robotic control [32, 3], where robot is initially designed symmetrical, but part of symmetry is destroyed by mechanical imperfection. Further examples of hidden symmetry can be also found in the literature on multi-agent reinforcement learning with a large number of agents. To avoid the curse of dimensionality, researchers often rely on the assumption of the existence of homogeneous agents [13, 34]. In the extreme case where all agents are homogeneous, such as in mean-field games, sample complexity becomes independent of the number of agents [13]. However, in practice, agents can be clustered into different types [34], and this information may not be known in advance to the learner (here symmetry occurs between different agents from the same type).

**Non-crossing partitions.** Sub-exponential size naturally appears when there is a hierarchical structure on the set $[d]$, and the partitioning needs to respect this hierarchical structure. Particularly, let $T(d, d_0)$ be the set of ordered trees with $(d + 1)$ nodes and $d_0$ internal nodes (i.e., nodes that are not the leaves). A partition that respects an ordered tree groups the children of the same node into a single equivalence class (for example, see Figure 1). It is shown in [15] that the cardinality of the set of partitions that respect ordered trees in $T(d, d_0)$ is sub-exponential. More precisely, it is $O(d^{2d_0})$. Furthermore, there is a natural bijection between partitions that respect ordered trees in $T(d, d_0)$ and the set of non-crossing partitions $\mathcal{NC}_{d,d_0}$ [15].

Recall the subcontractor example in the introduction. Here, after the company hires subcontractors $\{1, 4, 6\}$ to do the job, these subcontractors further break down the tasks into smaller subtasks and hire additional subcontractors $\{2, 3\}$, $\{5\}$, and $\{7, 8, 9\}$, respectively, to execute the subtasks.

**Non-nesting partitions.** Besides non-crossing partitions, another sub-exponential-size class of partitions with practical relevance is non-nesting partitions. Consider the resource allocation task where there are $d$ upcoming tasks and $d_0$ machines. The job of the designer is to allocate these tasks to each machine.

Now, assume each task will appear in time $t_1 < t_2 < \cdots < t_d$, but the exact time (the value of $t_i$) is unknown to the designer. Moreover, the cost of machine $k \in [d_0]$, given a subset of tasks $A_k$ (ordered according to execution time), is $c_k = t_{\max(A_k)} - t_{\min(A_k)}$.

The goal of the designer is to minimize the maximum cost of all machines:

$$\min \max_{k \in [d_0]} c_k.$$

To achieve this, the designer should avoid nesting allocations (i.e., searching among non-nesting partitions). In particular, assuming that if tasks at times $t_i$ and $t_j$ are assigned to machine $k$, (where $t_i < t_j$), and tasks at times $t_p$ and $t_q$ are assigned to machine $k'$ (where $t_p < t_q$), then it should not be the case that $t_i < t_p < t_q < t_j$. This is because the cost of machine $k$ would be significantly higher than that of machine $k'$, and the cost could be reduced by swapping task $t_q$ for machine $k$ with task $t_j$ for machine $k'$.

## D.3 Efficient Greedy Algorithm for Specific Classes of Partitions

The model selection procedure in the exploration phase of Algorithms 1 and 2 requires finding the best subspace in the pool $m \in \mathcal{M}$ with respect to least square errors. In the worst case, the algorithm needs to solve $M$ linear regression approaches. While the exact computation of the best subspace $\hat{m}$ as in (4) is an NP-hard problem in general (since it contains interval partition as a subclass), we argue that an greedy algorithm can find the ground-truth subspace $m_\star$ in $O(nd^5)$ time complexity, given sufficient large number of samples.

The pseudo code of the greedy algorithm is given in Figure 4. Given that the set of partitions $\mathcal{Q}_d$ is equipped with a lattice structure, in which the finest partition is $(1|2|\ldots|d)$ and the coarsest partition is $(1, 2, \ldots, d)$.

The algorithm starts with the finest partition $\hat{\pi} = (1|2|\ldots|d)$. In each iteration, the algorithm finds the finest coarsening of the current partition $\hat{\pi}$. In graph-theoretic terms, it finds all neighbors of $\hat{\pi}$ in

the lattice that are coarsenings of $\hat{\pi}$. The operator for finding the finest coarsening of $\hat{\pi}$ is denoted as `Coarsen`$(\hat{\pi})$, and it returns a collection of the finest coarsened partitions. Next, the algorithm finds the partition that minimizes the prediction error $\|Y - \Pi_{S_m}(Y)\|_2^2$ among the current coarsening collection. At the end of each iteration, as the number of classes in $\hat{\pi}$ is reduced by one, the dimension variable is also reduced by 1. The while loop stops when the dimension equals $d_0$.

Since the algorithm only optimizes locally within the current coarsening collection, it exhibits a behavior similar to a greedy algorithm. We note that for non-crossing and non-nesting partitions, the cardinality of the coarsening collection at any level of the lattice is at most $d^2$. Therefore, assuming that creating a finest coarsening partition take $O(d)$ operator, and solving least square takes $O(nd^2)$, the algorithms time complexity is $O(nd^5)$.

---

**Algorithm 4** Greedily Search within Lattice

1: Input: Compact representation of $\mathcal{M}$, design matrix $X$, reward vector $Y$.
2: Initialise $\hat{\pi} = (1|2|3|4...|d)$, dimension $= d$.
3: **while** dimension $> d_0$ **do**
4:     Collection $= $ `Coarsen`$(\hat{\pi})$.
5:     $\hat{m} = \arg\min_{m \in \mathbf{H}(\text{Collection})} \|Y - \Pi_{S_m}(Y)\|_2^2$.
6:     $\hat{\pi} = \mathbf{H}^{-1}(\hat{m})$.
7:     dimension $=$ dimension $- 1$.
8: **end while**
9: $\hat{\theta} = \arg\min_{\theta \in \hat{m}} \|Y - X\theta\|_2^2$.
10: Return $\hat{m}, \hat{\theta}$.

---

# E   Experiment details

We conduct simulations where the entries of $\theta_\star$ satisfy non-crossing partition constraints. The set of arms $\mathcal{X}$ is $\sqrt{d}\mathbb{S}^{d-1}$, $\sigma = 0.1$, and $(d, d_0) \in \{(40, 4), (80, 10), (100, 15)\}$. We let exploratory distribution $\nu$ be the uniform distribution on the unit sphere. The ground-truth partition $\pi_{\mathcal{G}}$ and $\theta_\star$ are randomized before each simulation.

To run ESTC-Lasso algorithm [23], we introduce an auxiliary sparse vector $\varphi$ corresponding to $\theta_\star$, whose entries are defined as $\varphi_i = \theta_{i+1} - \theta_i$, and $\varphi_d = \theta_d$. We apply Lasso regression for $\varphi_\star$, get the estimate $\hat{\varphi}$, then convert back to $\hat{\theta}$ using the map that transforms sparse vector to interval-partition vector (inversion of the map we defined above).

Regarding implementing our algorithm, we use greedy Algorithm 4 as introduced in Appendix D.3, to solve the optimisation in equation (3), (4), and its complexity is $O(t_1 d^5)$. It is shown in the simulation result (Figure 2, 3, 4), the greedy algorithm achieves small risk error and consequently leads to small regret. Code is available at:
`https://github.com/NamTranKekL/Symmetric-Linear-Bandit-with-Hidden-Symmetry.git`.

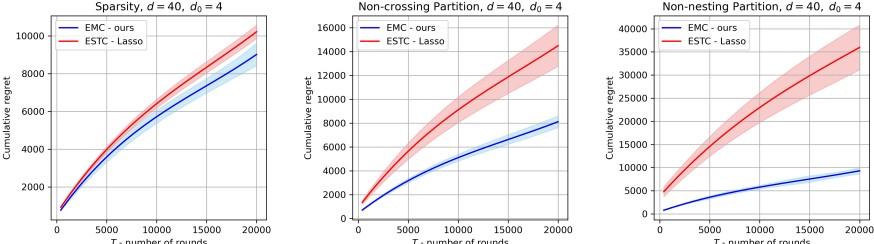

**Figure 3:** Regret of EMC (Algorithm 1) and of ESTC proposed in [23], in cases of sparsity, non-crossing partitions, and non-nesting partitions, with $d = 40$, $d_0 = 4$.

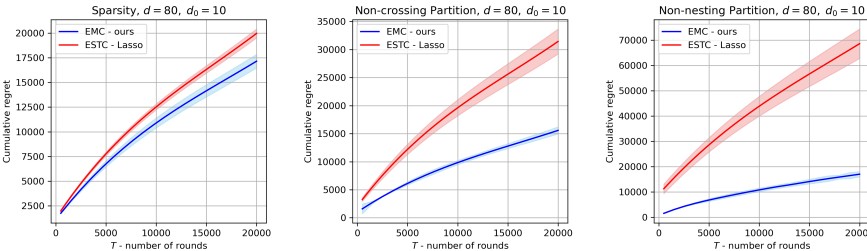

**Figure 4:** Regret of EMC (Algorithm 1) and of ESTC proposed in [23], in cases of sparsity, non-crossing partitions, and non-nesting partitions, with $d = 80$, $d_0 = 10$.

# F    Extended Related Work

**Sparse linear bandits.** As we will explain in Section 4.2, sparsity is equivalent to a subset symmetry structures, and thus, can be seen as a special case of our setting. As such, we first review the literature of sparsity. Sparse linear bandits were first investigated in [2], where the authors achieve a regret of $\tilde{O}(\sqrt{dsT})$, with $\tilde{O}$ disregarding the logarithmic factor, and $s$ representing the sparsity level, and $T$ is the time horizon. Without additional assumptions on the arm set and feature distribution, the lower bound for regret in the sparsity case is $\Omega(\sqrt{dsT})$ [27]. Consequently, the contextual setting has recently gained popularity in the sparsity literature, where additional assumptions are made regarding the context distribution and set of arms. With this assumption, it can be shown that one can achieve regret of the form $\tilde{O}(\tau s\sqrt{T})$, where $\tau$ is a problem-dependent constant that may have a complex form and varies from paper to paper [26, 36]. Apart from the contextual assumption, to avoid polynomial dependence on $T$ in the regret bound, assumptions are required for the set of arms [28, 11, 23]. Recently, [23] offers a unified perspective on the assumption regarding the set of arms by assuming the existence of an exploratory distribution on the set of arms. With this assumption, the authors propose an Explore then Commit style strategy that achieves $\tilde{O}(s^{\frac{2}{3}}T^{\frac{2}{3}})$, nearly matching the lower bound $\Omega(s^{\frac{2}{3}}T^{\frac{2}{3}})$ in the poor data regime [24]. As the sparsity structure can be reduced to a subset of the symmetry structure, all the lower bounds for sparse problems apply to (unknown) symmetric problems.

**Model selection.** Our problem is also closely related to model selection in linear bandits, as the learner can collect potential candidates for the unknown symmetry. Bandit model selection involves the problem where there is a collection of $M$ base algorithms (with unknown performance guarantees) and a master algorithm, aiming to perform as well as the best base algorithm. The majority of the literature assumes the black-box collection of models $M$ base algorithms and employs a variant of online mirror descent to select the recommendations of the base agent [4, 37, 38]. Due to the black-box nature, the regret guarantee bound depends on poly($M$). There is a growing literature on model selection in stochastic linear bandits, where there is a collection of $M$ features, and linear bandits running with these features serve as base algorithms. By exploiting the fact that the data can be shared across all the base algorithms, the dependence of regret in terms of the number of models can be reduced to $\log(M)$. In particular, [25] propose a method that concatenates all $M$ features of dimension $d$ into one feature of dimension $Md$, and then runs a group-Lasso bandit algorithm on top of this concatenated feature space, using the Lasso estimation as a aggregation of models. Their algorithm achieves a regret bound of $O(T^{\frac{3}{4}}\sqrt{\log(M)})$ under the assumption that the Euclidean norm of the concatenated feature is bounded by a constant. However, in our case, the Euclidean norm of concatenated feature can be as large as $\sqrt{M}$, which leads to $\sqrt{M}$ multiplicative factor in regret. Besides, [35] uses the online aggregation oracle approach, and able to obtain regret as $O(\sqrt{KdT\log(M)})$, where $K$ is the number of arms. In contrast, we use different algorithmic mechanism than aggregation of models. In particular, we explicitly exploiting the structure of the model class as a collection of subspaces and invoking results from Gaussian model selection [21] and dimension reduction on the union of subspaces [8]. With this technique, we are able to achieve $O(T^{\frac{2}{3}}\log(M))$, which is rate-optimal in the data-poor regime, has logarithmic dependence on $M$ without strong assumptions on the norm of concatenated features, and is independent of the number of arms $K$. A special case of feature selection where one can achieve a very tight regret compared to the best model is the nested feature class [18, 20]. In particular, in the nested feature class where

dimensions range from $\{1, \ldots, d\}$, and $d_{m_\star} < d$ represents the realizable feature of the smallest dimension, the regret bound can be $\tilde{O}(\sqrt{Td_{m_\star}})$ as shown in [20]. While the regret bound nearly matches the regret of the best model in the nested feature class, the assumption on nested features cannot be applied in our setting.

**Symmetry in online learning.** The notion of symmetry in Markov Decision Making dates back to works such as [22, 40]. Generally, the reward function and probability transition are preserved under an action of a group on the state-action space. Exploiting known symmetry has been shown to help achieve better performance empirically [43, 42] or tighter regret bounds theoretically [41]. However, all these works requires knowledge of symmetry group, while setting consider unknown subgroup which may be considerably harder. Unknown symmetry on the context or state space has been studied by few authors, with the term context-lumpable bandit [29], meaning that the set of contexts can be partitioned into classes of similar contexts. It is important to note that the symmetry group acts differently on the context space and the action space. As we shall explain in detail in Section 3, while one can achieve a reduction in terms of regret in the case of unknown symmetry acting on context spaces [29], this is not the case when the symmetry group acts on the action space. Particularly, we show that without any extra information on the partition structure of similar classes of arms, no algorithm can achieve any reduction in terms of regret.

The work closest to ours is [39], where the authors consider the setting of a $K$-armed bandit, where the set of arms can be partitioned into groups with similar mean rewards, such that each group has at least $q > 2$ arms. With the constrained partition, the instance-dependent regret bounds are shown asymptotically to be of order $O\left((K/q)\log T\right)$. Comparing to [39], we study the setting of stochastic linear bandits with similar arms, in which the (unknown) symmetry and linearity structure may intertwine, making the problem more sophisticated. We also impose different constraints on the way one partitions the set of arms, which is more natural in the setting of linear bandits with infinite arms. As a result, we argue that the technique used in [39] cannot be applied in our setting. To clarify this, let us first review the algorithmic technique from [39]: The algorithm in [39] assumes there is an equivalence among the parameters $\theta$, and that the set of arms $\mathcal{X}$ is a simplex. At each round $t$, given an estimation $\hat{\theta}_t$, the algorithm maintains a sorted list of indices in $[d]$ that follows the ascending order of the magnitude of $\hat{\theta}_i$. The algorithm then uses the sorted list $(\hat{\theta}_i)_{i\in[d]}$ to choose arm $x$ accordingly. The key assumption here is that, since the set of arms $\mathcal{X}$ is a simplex, we can estimate each $\theta_i$ independently. This implies that the list $(\hat{\theta}_i)_{i\in[d]}$ should respect the true order of the list $(\theta_i)_{i\in[d]}$ when there are a sufficiently large number of samples. Unfortunately, this is typically not the case in linear bandits where $\mathcal{X}$ has a more general shape. In linear bandits, there can be correlations between the estimates $\hat{\theta}_i$ and $\hat{\theta}_j$ for any $i, j \in [d]$. Hence, one should not expect that $(\hat{\theta}_i)_{i\in[d]}$ will maintain the same order as $(\theta_i)_{i\in[d]}$. In other words, the correlations among the estimates $\{\hat{\theta}_1, \ldots, \hat{\theta}_d\}$ may destroy the original order in $\{\theta_1, \ldots, \theta_d\}$. In fact, we can only guarantee the risk error of estimation $\theta$, i.e., $\|\hat{\theta} - \theta_\star\|$ is small, but not necessarily the order of the indices in $\theta$. Therefore, the technique used in [39] cannot be directly applied to linear bandits in its current form.

