# OpenReview forum: "Symmetric Linear Bandits with Hidden Symmetry"
_NeurIPS.cc/2024/Conference — NeurIPS 2024 poster_

### Official Review · Reviewer_ZVeC · 2024-07-07

**Soundness:** 3
**Presentation:** 3
**Contribution:** 4
**Rating:** 7
**Confidence:** 3

**Summary:**

The paper introduces and analyzes the problem of symmetric linear bandits with hidden symmetry. The authors study high-dimensional linear bandits where the reward function is invariant under certain unknown group actions on the set of arms. The key contributions are:

- An impossibility result showing that no algorithm can benefit solely from knowing that the symmetry group is a subgroup of permutation matrices, necessitating further structural assumptions.
- Establishment of a cardinality condition on the class of symmetric linear bandits with hidden symmetry, under which the learner can overcome the curse of dimensionality.
- Introduction of a new algorithm called EMC (Explore Models then Commit) that achieves a regret bound of O(d_0^(1/3) * T^(2/3) * log(d)), where d is the ambient dimension and d_0 is the dimension of the true low-dimensional subspace (d_0 << d).
- An improved regret bound of O(d_0 * sqrt(T) * log(d)) under an additional assumption of well-separated models.
- Discussion of an open problem regarding adaptive algorithms that can achieve optimal regret in both well-separated and general cases.

The paper provides theoretical analysis and proofs for the proposed algorithms and bounds. The authors position their work in the context of existing literature on sparse linear bandits and model selection, highlighting the novelty of their approach in leveraging symmetry for efficient exploration in high-dimensional linear bandits.

This paper is primarily theoretical, focusing on the mathematical formulation of the problem, algorithm design, and regret analysis. It does not include experimental results.

**Strengths:**

1. Originality:
   - Introduces a novel problem formulation of symmetric linear bandits with hidden symmetry, extending and generalizing the well-studied sparse linear bandit setting.
   - Creatively combines ideas from group theory, model selection, and bandit algorithms to address this new problem.
   - The proposed EMC algorithm represents an innovative approach to leveraging hidden symmetry in high-dimensional bandits.

2. Quality:
   - The technical analysis is rigorous and thorough, with well-structured proofs for all major claims.
   - Provides a comprehensive theoretical framework, including an impossibility result, regret bounds, and algorithm analysis.
   - Builds upon and extends existing results in a mathematically sound manner.

3. Clarity:
   - The problem motivation and significance are clearly articulated.
   - The paper is well-structured, with a logical flow from problem formulation to theoretical results.
   - Technical concepts are generally well-explained, with appropriate use of mathematical notation.
   - Some complex concepts, particularly around the equivalence between sparsity and interval partitions, could benefit from more intuitive explanations or concrete examples for broader accessibility.

4. Significance:
   - Addresses an important gap in the literature by considering hidden symmetry in high-dimensional linear bandits.
   - Provides new insights into the role of symmetry in sequential decision-making, with potential broad implications for reinforcement learning and online optimization.
   - Establishes connections between symmetry structures and sparsity, potentially opening new avenues for efficient exploration in high-dimensional spaces.
   - The proposed algorithms and bounds represent significant progress in overcoming the curse of dimensionality in certain bandit settings.

**Weaknesses:**

1. Limited empirical validation:
   - The paper appears to be primarily theoretical, with no mention of experimental results or simulations to validate the proposed algorithms.
   - Including empirical studies, even on synthetic datasets, would strengthen the practical relevance of the theoretical results.
   - Comparison with existing methods in terms of computational efficiency and performance could provide valuable insights.

2. Complexity of concepts:
   - Some key concepts, such as the equivalence between sparsity and interval partitions, are not explained intuitively enough for a broader audience.
   - Additional examples or visual representations could make these complex ideas more accessible.

3. Practical applicability:
   - The paper lacks a detailed discussion on how the proposed algorithms could be implemented in real-world scenarios.

4. Computational complexity:
   - There's limited discussion on the computational complexity of the proposed algorithms, particularly for the EMC algorithm.
   - Understanding the trade-offs between theoretical performance and computational requirements is crucial for practical implementation.

5. Assumptions and limitations:
   - While the paper acknowledges some limitations, a more comprehensive discussion of the assumptions' implications and potential violations in real-world scenarios would be beneficial.

6. Comparison with related approaches:
   - While the paper discusses how it differs from some existing methods, a more comprehensive comparison with other approaches to high-dimensional bandits or symmetry exploitation could provide better context.

These weaknesses are not meant to diminish the paper's overall contribution but rather to identify areas where the work could be further strengthened or extended in future research.

**Questions:**

Empirical Validation:
   Could you provide any empirical results, even on synthetic data, to illustrate the performance of the EMC algorithm? How does it compare practically to existing methods for sparse linear bandits?

Practical Implications of Assumptions:
   Could you elaborate on real-world scenarios where Assumption 5 (sub-exponential number of partitions) and Assumption 16 (well-separated partitioning) are likely to hold or be violated?

Intuitive Explanations for Key Concepts:
  The connection between interval partitions and sparsity, as well as the concept of hidden symmetry, are intriguing but complex. Could you provide more intuitive explanations or concrete examples to illustrate these key ideas, particularly for readers less familiar with group theory or symmetry concepts in bandits?

**Limitations:**

The authors have adequately addressed the limitations of their work, particularly by acknowledging open problems and discussing constraints of their assumptions. While they could expand on practical implementation challenges and potential societal impacts, their upfront approach to discussing limitations is commendable. A brief section explicitly addressing broader implications would further strengthen the paper, but overall, the authors have done a good job in addressing limitations within the context of their theoretical work.

---

> ### Author Rebuttal · Authors · 2024-08-06
>
> # Responses to Reviewer ZVeC
> We thank the reviewer for the insightful comments. Below are our responses/clarifications to your questions:
>
> ## Question 1: Practical Implications of Assumptions 5 and 16:
>
> ### Response:
> **Assumption 5: sub-exponential partitions.**
> Sub-exponential size naturally appears when there is a hierarchical structure on the set $[d]$, and the partitioning needs to respect this hierarchical structure.
> Particularly, let $T(d,d_0)$ be the set of ordered trees with $(d+1)$ nodes and $d_0$ internal nodes (i.e., nodes that are not the leaves).
> A partition that respects an ordered tree groups the children of the same node into a single equivalence class.
> We provide an example of such a partition in the PDF file (Figure 1).
> It is shown in [1] that the cardinality of the set of partitions that respect ordered trees in $T(d,d_0)$ is sub-exponential. More precisely, it's $O(d^{d_0})$.
> Furthermore, there is a bijection between partitions that respect ordered trees in $T(d,d_0)$ and the set of non-crossing partitions $\mathcal{NC}_{d,d_0}$ [1].
>
> **A linear bandit example**: To further illustrate the occurrence of such symmetry in a linear bandit problem, consider the following example:
> Suppose there are $d$ workers, and each worker $i$ can put $x_i \in [0,1]$ level of effort into the task.
> Hence, $x = [x_i]_{i\in [d]} \in \mathbb{R}^d$ is a vector that represents the effort of all workers.
> The performance of the whole team is measured by
> $$
> f(x) = \left <x,\theta\right>,
> $$
>
> where $\theta \in \mathbb{R}^d$, and each entry $\theta_i > 0$ represents the significance of worker $i$ to the success of the whole project.
> In other words, a higher $\theta_i$ implies that $x_i$ has more impact on the success of the project.
>
> Now, a new manager, who does not know $\theta$, employs a bandit algorithm to optimize the performance $f$.
> While she does not know $\theta$, she has prior knowledge that the skill levels of each worker in $[d]$ are hierarchical, meaning the significance of workers to the task can be represented as an ordered tree.
> This is expected in practice, as workers may come from different skill sets (e.g., developing, maintenance, testing) and varying skill levels (from senior to junior).
> We refer the reviewer to the PDF file (Figure 1) for an illustration of such a partition with respect to the ordered tree.
> Suppose she knows that there are at most $d_0$ equivalence classes in the partition.
> In that case, the number of partitions that respect the tree structures (i.e., can only group children of the same node into one equivalence class) must be at most $O(d^{d_0})$, due to the fact mentioned earlier.
>
> **Assumption 16: well-separated partitions.**
> For Assumption 16 to hold, there must be a significant differences among the classes in the partition. Let consider our incentivising workers example.
> The differences among classes occur among classes occur, for instance, when each group consists of specialists who excel in very specific skills, making the differences among the groups noticeable. As such, one can easily distinguish individuals in different groups, which is essentially the notion of a well-separated partition.
>
> ## Question 2: Intuitive Explanation for Key Concepts
>
> ### Response:
> To illustrate the equivalence between interval partitions and sparsity, let us consider the following example.
> Let $\theta \in \mathbb{R}^5$ where its entries satisfy a linear order, that is, $\theta_1 < \theta_2 < \cdots < \theta_5$. For example, let $\theta = [1, 1, 2, 3, 3]$, which has $d_0 = 3$ equivalent classes.
> Now, to define the corresponding sparse pattern, let $\varphi \in \mathbb{R}^4$ with entries defined as $\varphi_i = \theta_{i+1} - \theta_i$, and $\varphi_d$ = $\theta_d$. Hence, if $\theta = [1, 1, 2, 3, 3]$, then $\varphi = [0, 1, 1, 0, 3]$, so $\varphi$ is a $d_0 = 3$-sparse vector.
>
> The concept of hidden symmetry, intuitively speaking, means that the set of arms $\mathcal X$ contains several equivalence classes, each of which has the same expected reward. However, hidden symmetry implies that the learner does not know the equivalence classes in advance and must learn them through data sampling.
>
>
> ## Comment 1: On the computational complexity.
> ### Response:
> We believe that to develop computationally efficient algorithms for a particular partition, we need to fully exploit its structure, similar to how existing literature has exploited the structure of sparsity.
> However, this sacrifices the generality of the result, especially if our aim is to establish a general condition on partitions under which one can achieve regret that scales with the dimension of the fixed-point subspace $d_0$.
> As establishing this general condition is the primary concern of the paper, we did not focus on computational efficiency.
> We are aware that developing efficient computational methods is important in practice, and we hope to investigate this question in the future for some important classes of partitions other than sparsity, such as non-crossing partitions.
>
> Moreover, as mentioned in Remark 4 of the paper, since the prediction error for each model $m \in \mathcal M$ can be computed independently, we can exploit parallel computing to reduce the algorithm's computation time.
>
> ## Comment 2: Empirical validation.
> ### Response:
>
> We run the simulation, with $d = 16$, $d_0 = 2$, $\mathcal X$ is the unit ball, with two scenarios: interval partition, and non-crossing partition. Due to the space constraints, we refer the reviewer to the discussion with Reviewer Viv8 (our response to their Comment 3) for simulation details, and the attached PDF file (Figure 2, 3) for simulation results. The simulation results show that, our algorithm achieves similar (or even smaller) regret in the case of interval partitions, and notably smaller regret in the case of non-crossing partitions compared to the sparse bandit algorithm.
>
> ## References:
> [1] Dershowitz&Zaks. Ordered trees and non-crossing partitions. 1986.

---

> > ### Author Response · Authors · 2024-08-12
> >
> > Dear Reviewer,
> >
> > Thank you for your time and effort in reviewing our paper. We hope our responses have addressed your concerns and questions. If you have any further questions, please don’t hesitate to let us know.
> >
> > Best regards,
> >
> > The Authors

---

### Official Review · Reviewer_qJvB · 2024-07-08

**Soundness:** 3
**Presentation:** 2
**Contribution:** 2
**Rating:** 3
**Confidence:** 4

**Summary:**

The authors study the problem of symmetric linear bandits with hidden symmetry (where the expected reward is a linear function of the selected arm, and is invariant under a hidden symmetry group). They show that, with no additional information, the minimax regret cannot be improved. When the partition corresponding to the hidden symmetry group is known to belong to a given set of a sub-exponential number of partitions, the authors provide an algorithm that that achieves a regret of $\tilde{O}(d_0^{1/3} T^{2/3})$. Under the assumption that the models are well separated, this regret guarantee is improved to $\tilde{O}(d_0 \sqrt{T})$.

**Strengths:**

The authors generalize the study of sparse linear bandits to high-dimensional symmetric linear bandits, where the symmetry is unknown and must be learned. The authors show that 1) no algorithm can benefit solely from knowing that there exists some subgroup under which the expected reward is invariant, 2) that a regret of $\tilde{O}(d_0^{1/3} T^{2/3})$ can be achieved when the partition is known to belong to a set of sub-exponential size, and 3) that a regret of $\tilde{O}(d_0 \sqrt{T})$ can be achieved when the models are well separated. For the well separated case, they note that the initialization phase length $t_2$ depends on the separation $\epsilon_0$, and question how to adapt to this parameter that will be unknown in practice.

**Weaknesses:**

- Adaptivity: the algorithm requires as input the set $\mathcal{Q}_{d,\le d_0}$ (currently not shown as in input). Additionally, the algorithm / regret do not appear to adapt to the complexity of the problem instance, but rather depend on the size of this input set. The manuscript should be revised to clarify (in the algorithm) that this is needed as input.
- The algorithm does not appear to be implementable. The minimization step in equation 5 is over $\mathcal{M}$, which is of exponential size ($d^{d_0}$). In the high dimensional regime of interest, this is infeasible without additional structural assumptions. The authors touch on this at the very end of the paper (line 402: "for future work, we will explore convex relaxation techniques for efficient computation"), but do not provide any concrete suggestions for how this could be done.
- Practical motivation: the authors do not provide any concrete examples of where this problem arises, and how the input set $\mathcal{Q}_{d,\le d_0}$ could be obtained in practice. The paper should be self contained and self-motivated (e.g. Line 20, referencing [24] alone is insufficient). The illustrative example of an ant robot does not seem very related to the problem at hand, as there are at most 4 symmetries. The authors briefly discuss possible hidden symmetries in Appendix D, but don't provide examples where we should expect to see these symmetries (anything besides sparsity, for which there already exist specialized methods).
- Clarity: the paper is quite difficult to read, and should be revised for clarity. I've listed some typos below, but thorough proofreading is needed.


Writing typos:
- Line 9: hidden symmetry
- Line 16: "Stochastic bandit is" incomplete sentence
- Line 26: "fo"
- Line 38: "most of studies"
- Line 81: "the set of arm is exploratory". Also, exploratory is undefined (reused in line 98).
- Line 100: data-poor regime is undefined
- Line 112: "And able to obtain regret"
- Line 114: "That are different to aggregation"
- Line 120: "Making" shouldn't be capitalized
- Line 124: missing "the"
- Line 142: "*In* each round"
- Line 148: in term*s* of regret
- Line 179: grammar, missing "to" and "as"->"be"
- Line 204: in term*s* of regret
- Line 206: "the" missing before "regret"
- Line 221: "are"->"is", unnecessary "the"
- Line 222: partition*s*
- Line 320: "designed"->"the design"
- general comment: "the assumptions" -> "assumptions"

Math typos / comments:
- Line 144: Is $\eta_t$ supposed to be a 0 mean $\sigma$-sub-Gaussian random variable?
- Line 144: Clearer to write $f(x_t) = \langle x_t, \theta_{\star}\rangle$
  - Equations 1 and 2: $\phi$ and $\hat{\phi}$ perform the same operation on $\mathbb{R}^d$, so it is unclear why they are separately defined for operating on $x$ and on $\theta$. Also, $\hat{\phi}$ is confusing notation to use, as this is not an estimator of $\phi$.
- Line 163: missing $\forall g \in \mathcal{G}$
- Line 188: should explicitly define dim before usage
- Prop 3: some intuitive explanation / proof sketch would be helpful.
- Line 234: "data"->"actions"
- Algorithms 1 and 2 critically require as input $\mathcal{Q}_{d,\le d_0}$.

**Questions:**

See weaknesses. At a high level:
1. Adaptivity: the algorithm requires as input the set $\mathcal{Q}_{d,\le d_0}$ , and it is not clear how to obtain this in practice.  Additionally, it appears that the key regret improvement is the scaling with the log of the cardinality of this set, instead of with $d$. Is there a way to make these assumptions less restrictive, adapt to the ``true'' complexity of the partition, or to estimate this set in practice?
2. Implementability: the algorithm as written does not seem to be implementable. Can the authors either provide an implementation (comparing their results with existing algorithms for sparse linear bandits), or suggest how this could be done in practice?
3. Practical motivation: is there a motivating example (not sparsity) where the arms are high dimensional, the symmetry is unknown, and the partition is known to belong to a set of sub-exponential size?

**Limitations:**

The key limitations of this method, that have not been sufficiently discussed, are its required knowledge of $\mathcal{Q}_{d,\le d_0}$, and its computational intractability, both of which appear to be severe practical limitations.

---

> ### Author Rebuttal · Authors · 2024-08-06
>
> # Responses to Reviewer qJvB
>
> ## Question: Practical motivation on sub-exponential partitions
> ### Response:
> Sub-exponential size naturally appears when there is a hierarchical structure on the set $[d]$, and the partitioning needs to respect this hierarchical structure.
> Particularly, let $T(d,d_0)$ be the set of ordered trees with $(d+1)$ nodes and $d_0$ internal nodes (i.e., nodes that are not the leaves).
> A partition that respects an ordered tree groups the children of the same node into a single equivalence class.
> We provide an example of such a partition in the PDF file (Figure 1).
> It is shown in [1] that the cardinality of the set of partitions that respect ordered trees in $T(d,d_0)$ is sub-exponential. More precisely, it's $O(d^{d_0})$.
> Furthermore, there is a bijection between partitions that respect ordered trees in $T(d,d_0)$ and the set of non-crossing partitions $\mathcal{NC}_{d,d_0}$ [1].
>
> **A linear bandit example**: To further illustrate the occurrence of such symmetry in a linear bandit problem, consider the following example:
> Suppose there are $d$ workers, and each worker $i$ can put $x_i \in [0,1]$ level of effort into the task.
> Hence, $x = [x_i]_{i\in [d]} \in \mathbb{R}^d$ is a vector that represents the effort of all workers.
> The performance of the whole team is measured by
> $$
> f(x) = \left <x,\theta\right>,
> $$
>
> where $\theta \in \mathbb{R}^d$, and each entry $\theta_i > 0$ represents the significance of worker $i$ to the success of the whole project.
> In other words, a higher $\theta_i$ implies that $x_i$ has more impact on the success of the project.
>
> Now, a new manager, who does not know $\theta$, employs a bandit algorithm to optimize the performance $f$.
> While she does not know $\theta$, she has prior knowledge that the skill levels of each worker in $[d]$ are hierarchical, meaning the significance of workers to the task can be represented as an ordered tree.
> This is expected in practice, as workers may come from different skill sets (e.g., developing, maintenance, testing) and varying skill levels (from senior to junior).
> We refer the reviewer to the PDF file (Figure 1) for an illustration of such a partition with respect to the ordered tree.
> Suppose she knows that there are at most $d_0$ equivalence classes in the partition.
> In that case, the number of partitions that respect the tree structures (i.e., can only group children of the same node into one equivalence class) must be at most $O(d^{d_0})$, due to the fact mentioned earlier.
>
>  ## Question: adapt to the true complexity of the partition instead of using the cardinality, or to estimate this collection of partition in practice?
>
> ### Response:
> We thank the reviewer for the interesting suggestion.
> However, as the literature of partitions with constraints is diverse with many different types of partitions, such as noncrossing partitions [2], non-nesting partitions [3], pattern avoidance partitions [4], partitions with distance restrictions [5], it is highly non-trivial to find unifying parameters to measure the hardness of partitions in learning tasks.
> The question of finding the right measure of complexity for partitions as well as adapting to this complexity hardness is challenging, hence we leave it to future work.
> For the current paper, as the unifying measure of complexity hardness for partitions is still unclear, we have decided use the cardinality of the set as a natural measure of complexity, which is a reasonable choice.
>
> In particular, the complexity of partitions with an extremely rich structures, can also be measured by cardinality:
> Take an interval partition, for example.
> Although its lattice structure is rich, its complexity relevant to the learning task is still $d_0 \log(d)$, which is the same as $\log(|\mathcal{Q}_{d,\leq d_0}|)$.
> Therefore, we argue that cardinality is useful for learning with partitions, including those with rich structure.
>
> ## Comment: Computational limitation.
> ### Response:
> We believe that to develop computationally efficient algorithms for a particular partition, we need to fully exploit its structure, similar to how existing literature has exploited the structure of sparsity.
> However, this sacrifices the generality of the result, especially if our aim is to establish a general condition on partitions under which one can achieve regret that scales with the dimension of the fixed-point subspace $d_0$.
> As establishing this general condition is the primary concern of the paper, we did not focus on computational efficiency.
> We are aware that developing efficient computational methods is important in practice, and we hope to investigate this question in the future for some important classes of partitions other than sparsity, such as non-crossing partitions.
>
> Moreover, as mentioned in Remark 4 of the paper, since the prediction error for each model $m \in \mathcal M$ can be computed independently, we can exploit parallel computing to reduce the algorithm's computation time.
>
> ## Comment: Empirical validation.
> ### Response:
>
> We run the simulation, with $d = 16$, $d_0 = 2$, $\mathcal X$ is the unit ball, with two scenarios: interval partition, and non-crossing partition. Due to the space constraints, we refer the reviewer to the discussion with Reviewer Viv8 (our response to Comment 3) for simulation details, and the attached PDF file (Figures 2 & 3) for simulation results. The simulation results show that, our algorithm achieves similar (or even smaller) regret in the case of interval partitions, and notably smaller regret in the case of non-crossing partitions compared to the sparse bandit algorithm.
>
> ## References:
> [1] Dershowitz&Zaks. Ordered trees and non-crossing partitions. 1986.
>
> [2] Baumeister et al. Non-crossing partitions. 2019.
>
> [3] Chen et al. Crossings and nestings of matchings and partitions. 2006.
>
> [4] B. E. Sagan. Pattern avoidance in set partitions. 2010.
>
> [5] Chu&Wei. Set partitions with restrictions. 2008.

---

> > ### Comment · Reviewer_qJvB · 2024-08-08
> > **Rebuttal Response**
> >
> > Thanks for the detailed response.
> >
> > 1. **Motivating example:** this seems interesting, and definitely should be included in the paper. I think that there some issues with this that still need to be fleshed out (e.g. in a workplace, one would expect to have the organization chart, i.e. the tree, revealed), but this appears to be a concrete setting going beyond sparsity.
> >
> > 2. **Adaptivity:** This remains my primary concern, which does not appear to have been addressed. The algorithm critically depends on knowledge of the set **$\mathcal{Q}_{d,\le d_0}$** , and cannot be run without this as input.
> > The algorithm, as written, does not currently take this as input, and so doesn't work. Additionally, the proposed bandit algorithm cannot adapt to the actual difficulty of the problem: if the partition actually belongs to a much smaller class **$\mathcal{Q}_{d,\le \tilde{d}_0}$**  where **$\tilde{d}_0 \ll d_0$**, the algorithm complexity still depends on **$\mathcal{Q}_{d,\le d_0}$**, the set it was provided as input.
> >
> > 3. **Computational Limitations:** the ability to evaluate models in parallel does not get at the core of the issue here, which is that for even reasonable $d$ and $d_0$ the number of models that need to be evaluated is an extremely large polynomial. Even considering the toy example provided by the authors in Figure 1, for a $d=20$ dimensional feature vector per individual, this yields $O(d^{d_0}) \approx 160000$ models to evaluate. Without an efficient implementation for this algorithm proposed in *any* setting, and no concrete plans for how this can be achieved, it is unclear how this algorithm can be used.
> >
> > 4. **Empirical validation:** I think that this addition will greatly strengthen the paper, as it shows that the algorithm can work in practice on small examples. However, it seems as though this example is quite artificial; as in the point above, considering the example the authors provided in Figure 1, even for this toy example, $d_0=4$, while the authors restricted to simulating $d_0=2$.
> >
> > The addition of the motivating example beyond sparsity and the addition of toy numerical results strengthens this paper, but due to the algorithmically required input of $\mathcal{Q}_{d,\le d_0}$, lack of adaptivity (beyond just evaluating this subset of models as opposed to all models), and the computational complexity scaling with $d^{d_0}$, I retain my score of reject.

---

> > > ### Author Response · Authors · 2024-08-08
> > > **Responses to reviewer qJvB**
> > >
> > > Dear Reviewer.
> > > Thank you for your reply. We would like to comment on these issues.
> > >
> > > ### Adaptivity
> > > We thank the reviewer for the insightful suggestion. As mentioned above, to adapt to benign problem instances, one might need to exploit the specific structure of a class of partitions, which is not the primary focus of this paper at the moment. However, it is indeed an interesting direction, and we leave it for future work.
> > >
> > > ### Computational limit.
> > > For particular classes of sub-exponential partitions, such as non-crossing and non-nesting, we can exploit their lattice structures and use greedy search to find the partition that yields reasonably small prediction error. Despite the NP-hardness of the computational complexity, it is often the case in practice that greedy search performs effectively. Moreover, one does not need to enumerate the set $\mathcal Q_{d,\leq d_0}$ before hand, as many classes of partitions admit compact representation (e.g., see [6], chapter 3).
> > >
> > > We hope that our response help to clarify your concern. We would like to kindly ask you to reevaluate your score.
> > >
> > > ### References
> > > [6] B. Baumeister, K.-U. Bux, F. Götze, D. Kielak, and H. Krause. Non-crossing partitions. 2019.

---

> > > > ### Comment · Reviewer_qJvB · 2024-08-08
> > > >
> > > > This response does not address my concerns, and so my score remains unchanged.
> > > >
> > > > Regarding computational limitations; would it be possible to show that if equation (5) is solved approximately (e.g. via greedy search), then the optimal $\theta$ found in the next step would not yield too much worse performance, and the regret would still be similarly bounded? Even if the set *$\mathcal{Q}_{d,\le d_0}$* doesn't need to be explicitly enumerated before hand, all models *$m \in \mathcal{Q}_{d,\le d_0}$* currently need to be tested, so without a way to cut down the size of this set (e.g. via greedy search or some efficient approximation scheme) the complexity of *$O(d^{d_0})$* will still be prohibitive.

---

> > > > > ### Author Response · Authors · 2024-08-08
> > > > > **Responses to reviewer qJvB**
> > > > >
> > > > > We thank the reviewer for the insightful suggestion. It is indeed very interesting to investigate whether greedy algorithms can find an approximation to the optimisation problem in equation (5). However, this is highly non-trivial and outside the scope of this paper, but we hope to explore this question in future work.
> > > > >
> > > > > Best regards,
> > > > > The Authors

---

### Official Review · Reviewer_WDJx · 2024-07-10

**Soundness:** 3
**Presentation:** 3
**Contribution:** 3
**Rating:** 7
**Confidence:** 3

**Summary:**

The paper studies high-dimensional linear bandits that are invariant w.r.t. an *unknown* subgroup of coordinate permutations. The authors first show through a lower bound that further information about the structure of the hidden subgroup is required to achieve a dimension-independent regret bound. They then propose a subexponential cardinality constraint on the hidden subgroup, which is sufficient to avoid the dimension dependency. They specifically propose *Explore-Models-then-Commit*, which successfully avoids the worst-case dimension-dependency under certain conditions.

**Strengths:**

- Well-written
- Clear motivation and a novel problem-setting of importance
- Solid motivation for subexponential cardinality assumption on the hidden subgroup, as well as interesting combinatorial concepts intertwined throughout the paper
- First good regret bounds in the case of hidden symmetry

**Weaknesses:**

- The only symmetry somewhat intuitive to me is the sparsity, equivalent to interval partitions. Despite the Introduction stating the importance of symmetry, it is unclear whether there is practically meaningful coordinate symmetry beyond sparsity.
(Of course, theoretically, I appreciate the results here.)

- No experimental results. Especially as the authors have stated that Algorithm 1 can be "parallelised using tools such as Ray", I was expecting at least some toy experiments (one showing the efficacy of learning the hidden symmetry and scaling of the algorithm as the number of models $M$ increase)

- The algorithm is explore-then-commit style, which requires the horizon length $T$ in advance and inherits its suboptimality [1].



[1] https://papers.nips.cc/paper_files/paper/2016/hash/ef575e8837d065a1683c022d2077d342-Abstract.html

**Questions:**

- The paper states that as sparsity is equivalent to interval partitions, the lower bound of Hao et al. (2020) also trivially applies. Then, is this lower bound tight for other combinatorial structures with similar subexponential cardinality constraints, e.g., non-crossing partition?
- In sparse linear bandits, there is always an assumption about the context distribution or the arm set (e.g., compatibility condition, restricted eigenvalue, etc.). Can these assumptions be interpreted as part of the paper's proposed group-theoretic framework as well?
- (minor) Would the principles here be extendable to information-directed sampling [3]?


[2] https://proceedings.neurips.cc/paper/2020/hash/7a006957be65e608e863301eb98e1808-Abstract.html

[3] https://openreview.net/forum?id=syIj5ggwCYJ

**Limitations:**

Yes

---

> ### Author Rebuttal · Authors · 2024-08-06
>
> # Response to Reviewer WDJx
> We thank the reviewer for the insightful comments. Below are our responses/clarifications to your questions:
> ## Comment 1: Practical motivations
> ### Response:
> Sub-exponential size naturally appears when there is a hierarchical structure on the set $[d]$, and the partitioning needs to respect this hierarchical structure.
> Particularly, let $T(d,d_0)$ be the set of ordered trees with $(d+1)$ nodes and $d_0$ internal nodes (i.e., nodes that are not the leaves).
> A partition that respects an ordered tree groups the children of the same node into a single equivalence class.
> We provide an example of such a partition in the PDF file (Figure 1).
> It is shown in [1] that the cardinality of the set of partitions that respect ordered trees in $T(d,d_0)$ is sub-exponential. More precisely, it's $O(d^{d_0})$.
> Furthermore, there is a bijection between partitions that respect ordered trees in $T(d,d_0)$ and the set of non-crossing partitions $\mathcal{NC}_{d,d_0}$ [1].
>
> **A linear bandit example**: To further illustrate the occurrence of such symmetry in a linear bandit problem, consider the following example:
> Suppose there are $d$ workers, and each worker $i$ can put $x_i \in [0,1]$ level of effort into the task.
> Hence, $x = [x_i]_{i\in [d]} \in \mathbb{R}^d$ is a vector that represents the effort of all workers.
> The performance of the whole team is measured by
> $$
> f(x) = \left <x,\theta\right>,
> $$
>
> where $\theta \in \mathbb{R}^d$, and each entry $\theta_i > 0$ represents the significance of worker $i$ to the success of the whole project.
> In other words, a higher $\theta_i$ implies that $x_i$ has more impact on the success of the project.
>
> Now, a new manager, who does not know $\theta$, employs a bandit algorithm to optimize the performance $f$.
> While she does not know $\theta$, she has prior knowledge that the skill levels of each worker in $[d]$ are hierarchical, meaning the significance of workers to the task can be represented as an ordered tree.
> This is expected in practice, as workers may come from different skill sets (e.g., developing, maintenance, testing) and varying skill levels (from senior to junior).
> We refer the reviewer to the PDF file (Figure 1) for an illustration of such a partition with respect to the ordered tree.
> Suppose she knows that there are at most $d_0$ equivalence classes in the partition.
> In that case, the number of partitions that respect the tree structures (i.e., can only group children of the same node into one equivalence class) must be at most $O(d^{d_0})$, due to the fact mentioned earlier.
>
>
> ## Question 1: Is this lower bound tight for other combinatorial structures with similar subexponential cardinality constraints, e.g., non-crossing partition?
>
> ### Response:
> The lower bound derived from the sparsity case [2] applies to any class of partitions that includes interval partitions, such as non-crossing partitions [3] and non-nesting partitions [4], and thus is tight in these settings.
> However, this lower bound does not hold for smaller classes that do not contain a structure equivalent to interval partitions. It is still unknown what a tight lower bound would be in this case, and thus, remains as future work.
>
> ## Question 2: Can assumptions such as compatibility condition, restricted eigenvalue, be interpreted as part of the paper's proposed group-theoretic framework as well?
>
> ### Response:
> We note that many conditions, such as the compatibility condition and restricted eigenvalue condition, are carefully tailored to exploit specific structures of sparsity (e.g., interval partitions) [5].
> Therefore, we believe that developing such conditions for symmetric bandits requires exploiting a specific combinatorial structure, such as non-crossing partitions.
> Although this is beyond the scope of our paper, as we study a wide class of partitions, it is indeed an interesting question that we hope to investigate in future work.
>
> ## Question 3: Would the principles here be extendable to information-directed sampling?
>
> ### Response:
> Extending our framework to information-directed sampling is indeed a very interesting and challenging problem.
> Since bounding the information ratio as in [6] requires strongly exploiting the particular structure of sparsity, we conjecture that studying specific partitions to fully exploit their combinatorial structures is necessary to derive the information ratio bound.
> This remains an open question, which we leave for future work.
>
> ## Comment 2: Empirical validation.
> ### Response:
>
> We run the simulation, with $d = 16$, $d_0 = 2$, $\mathcal X$ is the unit ball, with two scenarios: interval partition, and non-crossing partition. Due to the space constraints, we refer the reviewer to the discussion with Reviewer Viv8 (our response to their Comment 3: Empirical validation) for simulation details, and the attached PDF file (Figure 2, 3) for simulation results. The simulation results show that, our algorithm achieves similar (or even smaller) regret in the case of interval partitions, and notably smaller regret in the case of non-crossing partitions compared to the sparse bandit algorithm.
>
> ## References:
> [1] Dershowitz&Zaks. Ordered trees and non-crossing partitions. 1986.
>
> [2] Hao et al. High-dimensional sparse linear bandits. 2020.
>
> [3] Baumeister et al. Non-crossing partitions. 2019.
>
> [4] Chen et al. Crossings and nestings of matchings and partitions. 2006.
>
> [5] S. A. V. de Geer and P. Bühlmann. On the conditions used to prove oracle results for the lasso. 2009.
>
> [6] Hao et al. Information directed sampling for sparse linear bandits. 2021.

---

> > ### Comment · Reviewer_WDJx · 2024-08-12
> >
> > Thank you for the responses, and apologies for getting back so late.
> >
> > After reading through the responses to my and other reviewer's reviews, I'm satisfied with the authors' responses and intend to keep my score.

---

### Official Review · Reviewer_Viv8 · 2024-07-11

**Soundness:** 4
**Presentation:** 3
**Contribution:** 3
**Rating:** 7
**Confidence:** 3

**Summary:**

The paper explores the impact of unknown symmetry on the regret for stochastic linear bandits. Under some assumptions on the set partition induced by the unknown subgroup G, the paper develops an "Explore-then-commit" algorithm that attains optimal scaling of the regret in terms of the dimension d_0 of the low-dimensional space induced by the group action.

**Strengths:**

* The paper describes the assumptions clearly and develops a regret bound that matches the lower bound for sparse linear bandits.
* The paper makes a good case motivating the generality of the symmetric bandits structure by showing how it can recover sparse bandits.
* A discussion comparing the results to those in the model aggregation literature is given.

**Weaknesses:**

- The writing could be made clearer in some sections. For eg, the implication $g \cdot \theta_* = \theta_*$ in line 164 seems valid only under some conditions on the set $\mathcal{X}$. Suppose $g$ only swaps the first and second components and all vectors in $\mathcal{X}$ have the same first and second components. It would be helpful if an example is used to describe the orbit and associated partition, the fixed-point subspace, etc.
- The implications of the assumption on $\pi_{\mathcal{G}}$ in line 218 and Assumption 5 could be described more clearly. Again, a short example might help.
- Experimental evaluation of their proposed approach is missing, Even a small simulation experiment could assure reader of the applicability of the algorithm.

**Questions:**

Please see weaknesses above

---

> ### Author Rebuttal · Authors · 2024-08-06
>
> # Responses to Reviewer Viv8
> We thank the reviewer for the insightful comments. Below are our responses/clarifications to your questions:
>
> ## Comment 1: On the condition of $\mathcal X$ so that $\theta_\star = g\cdot \theta_\star$.
>
> ### Response:
>
> We do not need to impose any condition on $\mathcal{X}$ for $\theta_\star = g\cdot \theta_\star$ to hold.
> In particular, we only require $f(g\cdot x) = f(x)$, but we do not require that the orbit under the action of $\mathcal{G}$ stays in $\mathcal X$, that is, we do not require, $g \cdot x \in \mathcal{X}$, for all $g \in \mathcal{G}$.
> Therefore, the condition $\theta_\star = g \cdot \theta_\star$ holds regardless of the shape of $\mathcal{X}$.
>
>
> ## Comment 2: On illustrative examples of partitions.
> ### Response:
> Sub-exponential size naturally appears when there is a hierarchical structure on the set $[d]$, and the partitioning needs to respect this hierarchical structure.
> Particularly, let $T(d,d_0)$ be the set of ordered trees with $(d+1)$ nodes and $d_0$ internal nodes (i.e., nodes that are not the leaves).
> A partition that respects an ordered tree groups the children of the same node into a single equivalence class.
> We provide an example of such a partition in the PDF file (Figure 1).
> It is shown in [1] that the cardinality of the set of partitions that respect ordered trees in $T(d,d_0)$ is sub-exponential. More precisely, it's $O(d^{d_0})$.
> Furthermore, there is a bijection between partitions that respect ordered trees in $T(d,d_0)$ and the set of non-crossing partitions $\mathcal{NC}_{d,d_0}$ [1].
>
> **A linear bandit example**: To further illustrate the occurrence of such symmetry in a linear bandit problem, consider the following example:
> Suppose there are $d$ workers, and each worker $i$ can put $x_i \in [0,1]$ level of effort into the task.
> Hence, $x = [x_i]_{i\in [d]} \in \mathbb{R}^d$ is a vector that represents the effort of all workers.
> The performance of the whole team is measured by
> $$
> f(x) = \left <x,\theta\right>,
> $$
>
> where $\theta \in \mathbb{R}^d$, and each entry $\theta_i > 0$ represents the significance of worker $i$ to the success of the whole project.
> In other words, a higher $\theta_i$ implies that $x_i$ has more impact on the success of the project.
>
> Now, a new manager, who does not know $\theta$, employs a bandit algorithm to optimize the performance $f$.
> While she does not know $\theta$, she has prior knowledge that the skill levels of each worker in $[d]$ are hierarchical, meaning the significance of workers to the task can be represented as an ordered tree.
> This is expected in practice, as workers may come from different skill sets (e.g., developing, maintenance, testing) and varying skill levels (from senior to junior).
> We refer the reviewer to the PDF file (Figure 1) for an illustration of such a partition with respect to the ordered tree.
> Suppose she knows that there are at most $d_0$ equivalence classes in the partition.
> In that case, the number of partitions that respect the tree structures (i.e., can only group children of the same node into one equivalence class) must be at most $O(d^{d_0})$, due to the fact mentioned earlier.
>
> ## Comment 3: Empirical Validation.
> ### Response:
>
> We run the simulation, with $d = 16$, $d_0 = 2$, $\mathcal X$ is the unit ball, with two scenarios:
>
> **Interval partition (i.e., sparse linear bandits)**:
>
> We first run our algorithm with the following $\theta$ whose entries satisfy interval partition constraints (i.e., it represents a sparse linear bandit setting):
> $$
> \theta_\star = [1,1,2,2...,2]
> $$
> Equivalently, we can introduce a sparse vector $\varphi$ corresponding to $\theta_\star$, defined as entries defined as $\varphi_i = \theta_{i+1} - \theta_i$, and $\varphi_d = \theta_d$. We have that $\varphi_\star = [0,1,0,...,0,2]$, which is $2$-sparse vector. We apply Lasso regression for $\varphi_\star$, get the estimate $\hat \varphi$, then convert back to $\hat \theta$ using the map that transforms sparse vector to interval-partition vector (inversion of the map we defined above). Then, we compare the regret of our algorithm with that of state-of-the-art sparse linear bandit algorithm introduced in [2].
> We show the regret bound of both algorithms in the attached PDF (Figure 2).
> It can be seen that our algorithm achieves similar regret (or even smaller) compared to the sparse bandit algorithm.
>
> **Non-crossing partition**:
>
> Now we run our algorithm with an $\theta$ whose entries satisfy non-crossing partition constraints but not interval partition constraints.
> $$\theta_\star = [1,1,2,2,2,1,1...,1].$$
> We use the same map to convert $\theta_\star$ to sparse vector $\varphi_\star$, run Lasso to get estimation $\hat \varphi$ and then convert back to get estimation $\hat \theta$. Then, we compare the regret of our algorithm with that of [2].
> We show the regret bound of both algorithms in the attached PDF (Figure 3).
> We can see from the plot, the regret of our algorithm is notably smaller than that of [2].
> It indicates that our algorithm performs better than Lasso-based algorithms (which is designed only for interval partition) in the general case of non-crossing partition.
>
> ## References:
> [1] Dershowitz&Zaks. Ordered trees and non-crossing partitions. 1986.
>
> [2] Hao et al. High-dimensional sparse linear bandits. 2020.

---

> > ### Author Response · Authors · 2024-08-12
> >
> > Dear Reviewer,
> >
> > Thank you for your time and effort in reviewing our paper. We hope our responses have addressed your concerns and questions. If you have any further questions, please don’t hesitate to let us know.
> >
> > Best regards,
> >
> > The Authors

---

### Official Review · Reviewer_qZUs · 2024-07-19

**Soundness:** 3
**Presentation:** 4
**Contribution:** 3
**Rating:** 6
**Confidence:** 3

**Summary:**

This paper studies stochastic linear bandits with hidden symmetry, where the reward function is invariant with respect to a subgroup $\mathcal{G}$ of coordinate permutations. The paper first presents an impossibility result, showing that solely knowing a low-dimensional symmetry structure exists does not help. However, when the collection of fixed-point subspaces of $\mathcal{G}$ is not too large, one can use model selection algorithms to first learn the symmetry structure. The paper provides regret bounds of the algorithms, including an improved bound under an additional assumption that the equivalence classes of coordinates are well-separated.

**Strengths:**

- The low-dimensional symmetry structure in linear bandits studied in this paper seems novel and interesting. It also generalizes the sparsity assumption that has often been considered in the literature.

- The impossibility result is not surprising but still good to have.

- The mathematical formulation is clean, and the connection between fixed-point subspaces and set partitions allows for a relatively simple notion of cardinality assumption for the algorithms to work.

- The improved regret bound in Section 5 leads to some interesting questions about additional structure that a learner could exploit.

- The technical results are clear, and the paper is well-written and easy to follow.

**Weaknesses:**

- The algorithms are not computationally-efficient.

- The algorithms require a rather strong assumption on the size of the collection of fixed-point subspaces of $\mathcal{G}$.

- While hidden symmetry is observed in many learning tasks (e.g., control, multi-agent reinforcement learning), the paper does not provide specific real-world applications of symmetry structures within the context of linear bandits.

- The paper can be much stronger with an empirical validation on at least synthetic data. In particular, it would be interesting to see how the algorithms behave on sparse linear bandits, in comparison with specialized algorithms.

**Questions:**

- In proposition 3, is the claim "let $\mathcal{G}$ be an unknown subgroup with $\dim(\mathrm{Fix}_{\mathcal{G}}) = 2$.

Then, for any algorithm, there exists $\theta^* \in \mathrm{Fix}_{\mathcal{G}}$ such that..."?

- What can be done if Assumption 5 does not hold?

- Can you elaborate on the comparison with [1], especially given that the symmetry structure here is on the coordinates so there are more similarities to the "groups of similar arms" assumption in multi-armed bandits? Could you further compare the settings and the techniques?

- In terms of presentation, I think a concrete running example in Section 2 can be very helpful.

[1] F. Pesquerel, H. Saber, and O. A. Maillard. Stochastic bandits with groups of similar arms. In Advances in Neural Information Processing Systems, 2021.

**Limitations:**

Assumptions are properly discussed.

---

> ### Author Rebuttal · Authors · 2024-08-06
>
> # Responses to  Reviewer qZUs
> We thank the reviewer for the insightful comments. Below are our responses/clarifications to your questions:
>
> ## Question 1: What can be done if Assumption 5 does not hold?
> ### Response:
> If the cardinality in Assumption 5 is not satisfied, one might need to find an alternative notion of *complexity* for a partition.
> However, as the literature of partitions with constraints is diverse with many different types of partitions, such as non-crossing partitions [1], non-nesting partitions [2], pattern avoidance partitions [3], partitions with distance restrictions [4], it is highly non-trivial to find unifying parameters to measure the complexity of partitions in learning tasks.
> As the unifying measure of complexity for partitions is unclear, one may study specific types of partitions to exploit their special properties and structures, such as the literature on sparsity (i.e., interval partitions), but this approach may lead to a loss of generality.
>
> ## Comment 1: Compare with [5], where symmetry appear in parameter space instead.
>
> ### Response:
>
> Let us first review the algorithmic technique of [5]:
> The algorithm assumes there is an equivalence among the parameters $\theta$, and that the set of arms $\mathcal{X}$ is a simplex. At each round $t$, given an estimation $\hat \theta_t$, the algorithm maintains a sorted list of indices in $[d]$ that follows the ascending order of the magnitude of $\hat \theta_i$.
> The algorithm then uses the sorted list of $ \hat{\theta}_i $ for all $i \in d$,  to choose arm $x$ accordingly.
> The key assumption here is that, since the set of arms $\mathcal{X}$ is a simplex, we can estimate each $\theta_i$ independently. This implies that the order of entries of $ \hat{\theta}$ , should respect the true order of the entries of $\theta$ when there are a sufficiently large number of samples.
>
> Unfortunately, this is typically not the case in linear bandits where $\mathcal{X}$ has a more general shape.
> In linear bandits, there can be correlations between the estimates $\hat \theta_i$ and $\hat \theta_j$ for any $i, j \in [d]$. Hence, one should not expect that the list of entries of  $\hat \theta $ will maintain the same order as that of $\theta$. In other words, the correlations among the estimates $\\{\hat \theta_1, \ldots, \hat \theta_d\\}$ may destroy the original order in $\\{\theta_1, \ldots, \theta_d\\}$.
> In fact, we can only guarantee the risk error of estimation $\hat \theta$, i.e., $\\| \hat \theta - \theta_\star \\|_2$ is small, but not necessarily the order of the indices in $\theta$.
> Therefore, the technique used in [5] cannot be directly applied to our setting in its current form.
>
> ## Comment 2: On the sub-exponential assumption, and practical examples
> ### Response:
> Sub-exponential size naturally appears when there is a hierarchical structure on the set $[d]$, and the partitioning needs to respect this hierarchical structure.
> Particularly, let $T(d,d_0)$ be the set of ordered trees with $(d+1)$ nodes and $d_0$ internal nodes (i.e., nodes that are not the leaves).
> A partition that respects an ordered tree groups the children of the same node into a single equivalence class.
> We provide an example of such a partition in the PDF file (Figure 1).
> It is shown in [6] that the cardinality of the set of partitions that respect ordered trees in $T(d,d_0)$ is sub-exponential. More precisely, it's $O(d^{d_0})$.
> Furthermore, there is a bijection between partitions that respect ordered trees in $T(d,d_0)$ and the set of non-crossing partitions $\mathcal{NC}_{d,d_0}$ [6].
>
> Due to space limitations, we refer the Reviewer to the discussion with Reviewer Viv8 (in our response to their Comment 2 - examples for partitions) and to the attached PDF file (Figure 1) for a specific example of a linear bandit with a collection of sub-exponential partitions.
>
> ## Comment 3: On the computational complexity.
> ### Response:
> We believe that to develop computationally efficient algorithms for a particular partition, we need to fully exploit its structure, similar to how existing literature has exploited the structure of sparsity.
> However, this sacrifices the generality of the result, especially if our aim is to establish a general condition on partitions under which one can achieve regret that scales with the dimension of the fixed-point subspace $d_0$.
> As establishing this general condition is the primary concern of the paper, we did not focus on computational efficiency.
> We are aware that developing efficient computational methods is important in practice, and we hope to investigate this question in the future for some important classes of partitions other than sparsity, such as non-crossing partitions.
>
> Moreover, as mentioned in Remark 4 of the paper, since the prediction error for each model $m \in \mathcal M$ can be computed independently, we can exploit parallel computing to reduce the algorithm's computation time.
>
> ## Comment 4: Empirical validation.
> ### Response:
>
> We run the simulation, with $d = 16$, $d_0 = 2$, $\mathcal X$ is the unit ball, with two scenarios: interval partition, and non-crossing partition. Due to the space limitations, we refer the reviewer to the discussion with Reviewer Viv8 (our response to their Comment 3) for simulation details, and the attached PDF file (Figure 2, 3) for simulation results. The simulation results show that, our algorithm achieves similar (or even smaller) regret in the case of interval partitions, and notably smaller regret in the case of non-crossing partitions compared to the sparse bandit algorithm.
>
> ## References:
>
> [1] Baumeister et al. Non-crossing partitions. 2019.
>
> [2] Chen et al. Crossings and nestings of matchings and partitions. 2006.
>
> [3] B. E. Sagan. Pattern avoidance in set partitions. 2010.
>
> [4] Chu&Wei. Set partitions with restrictions. 2008.
>
> [5] Pesquerel et al. Stochastic bandits with groups of similar arms. 2021.
>
> [6] Dershowitz&Zaks. Ordered trees and non-crossing partitions. 1986.

---

> > ### Author Response · Authors · 2024-08-12
> >
> > Dear Reviewer,
> >
> > Thank you for your time and effort in reviewing our paper. We hope our responses have addressed your concerns and questions. If you have any further questions, please don’t hesitate to let us know.
> >
> > Best regards,
> >
> > The Authors

---

> > ### Comment · Reviewer_qZUs · 2024-08-12
> >
> > Thank you for the detailed responses. I will maintain my score for now.

---

### Author Rebuttal · Authors · 2024-08-06

Thank you for your valuable and constructive feedback. We have performed the additional experiments as requested by the reviewers and have provided the results in this PDF file. We have also added a picture in this PDF file that illustrates a practical example of how sub-exponential partitioning occurs when there is a hierarchical structure on the set $[d]$.

---

### Decision · Program_Chairs · 2024-09-25

**Decision:**

Accept (poster)

**Comment:**

The paper focuses on a novel low-dimensional structure, exploitable/non-exploitable, in high-dimensional stochastic linear bandits. The paper asserts that the knowledge of the symmetry group (symmetric structure) as a subgroup of permutation matrices structures is not sufficient to achieve sublinear regret (impossibility result). The authors further demonstrate sublinear/minimax optimal regret guarantees under certain symmetries (stronger than permutation but beyond sparsity, a previously known symmetric structure). The guarantees match the minimax lower bound shown in the literature of sparse linear bandit, which is a significant finding acknowledged by some reviewers. The main contribution is theoretical, with practical implementation posing challenges in the proposed settings/algorithms, as commented by reviewers.

There seems to be a concern raised by a reviewer about the necessity of the input $\mathcal{Q}_{d, \le d_0}$. The discussion revealed the requirement of a special structure beyond the partition in the absence of this input, which would be better explained in the paper. There were also comments from several reviewers about the lack of real-world examples where symmetry becomes important. The authors have provided motivating examples that have been acknowledged by the reviewers. These should be included in the revised version of the paper.